# A low Smc flux avoids collisions and facilitates chromosome organization in *Bacillus subtilis*

Anna Anchimiuk[1], Virginia S Lioy[2], Florian Patrick Bock[1], Anita Minnen[3], Frederic Boccard[2], Stephan Gruber[1]*

[1]Department of Fundamental Microbiology, Faculty of Biology and Medicine, University of Lausanne, Lausanne, Switzerland; [2]Université Paris-Saclay, CEA, CNRS, Institute for Integrative Biology of the Cell (I2BC), Gif-sur-Yvette, France; [3]Max Planck Institute of Biochemistry, Martinsried, Germany

**Abstract** SMC complexes are widely conserved ATP-powered DNA-loop-extrusion motors indispensable for organizing and faithfully segregating chromosomes. How SMC complexes translocate along DNA for loop extrusion and what happens when two complexes meet on the same DNA molecule is largely unknown. Revealing the origins and the consequences of SMC encounters is crucial for understanding the folding process not only of bacterial, but also of eukaryotic chromosomes. Here, we uncover several factors that influence bacterial chromosome organization by modulating the probability of such clashes. These factors include the number, the strength, and the distribution of Smc loading sites, the residency time on the chromosome, the translocation rate, and the cellular abundance of Smc complexes. By studying various mutants, we show that these parameters are fine-tuned to reduce the frequency of encounters between Smc complexes, presumably as a risk mitigation strategy. Mild perturbations hamper chromosome organization by causing Smc collisions, implying that the cellular capacity to resolve them is limited. Altogether, we identify mechanisms that help to avoid Smc collisions and their resolution by Smc traversal or other potentially risky molecular transactions.

*For correspondence:
stephan.gruber@unil.ch

**Competing interests:** The authors declare that no competing interests exist.

## Introduction

Members of the family of SMC proteins are ubiquitous in eukaryotes and also present in most bacteria and at least some lineages of archaea. They are crucial for establishing 3D genome organization inside cells, laying the foundation for faithful segregation, recombination, and repair of the chromosomal DNA molecules. Together with kleisin and kite subunits (or kleisin and hawk subunits), SMC proteins form ATP-hydrolyzing DNA motors that actively fold chromosomal DNA molecules apparently by DNA loop extrusion (*Yatskevich et al., 2019*). Loop extrusion can explain diverse folding phenomena across all domains of life: formation of topologically associated domains (TADs) in interphase, lengthwise compacted chromosomes during mitosis, as well as juxtaposition of the arms of bacterial chromosomes.

Recently, ATP-dependent loop extrusion has been recorded in single-molecule experiments. Purified yeast condensin and vertebrate cohesin extrude DNA loops at rates of ~1 kb/s in an asymmetric (one-sided) or symmetric (two-sided) manner, respectively (*Davidson et al., 2019*; *Ganji et al., 2018*; *Kim et al., 2019*). Nevertheless, the molecular underpinnings of loop extrusion are yet to be discovered. In the case of yeast condensin, two DNA-loop-extruding complexes on the same DNA molecule were reported to occasionally traverse one another in vitro, thus forming interlocking loops (also termed Z loops) (*Kim et al., 2020*). In principle, the latter behavior could improve the otherwise poor loop coverage achieved by one-sided motors, but on the other hand it likely generates

undesirable DNA entanglements (such as pseudoknots) (*Banigan et al., 2020*). The biological relevance of Z-loop formation and condensin traversal is yet to be determined.

Two distinct patterns of chromosome organization have been described for bacteria. One relies on MukBEF (or MksBEF) complexes, presumably starting DNA loop extrusion from randomly chosen entry sites on the bacterial chromosome (*Lioy et al., 2018*; *Lioy et al., 2020*; *Mäkelä and Sherratt, 2020*). The second pattern occurs in most bacteria, where Smc-ScpAB complexes start DNA translocation from predefined entry sites, the 16 bp *parS* DNA sequences, which are generally found near the replication origin and are specifically recognized by the Smc-loader protein ParB (*Gruber and Errington, 2009*; *Lin and Grossman, 1998*; *Minnen et al., 2016*; *Sullivan et al., 2009*). ParB dimers form DNA clamps that self-load onto DNA at *parS* sites (*Jalal et al., 2020*; *Osorio-Valeriano et al., 2019*; *Soh et al., 2019*). As Smc complexes translocate away from the *parS* loading site in both directions (two-sided), they co-align the left and the right chromosome arms that flank the replication origin (*Marbouty et al., 2015*; *Minnen et al., 2016*; *Tran et al., 2017*; *Wang et al., 2015*; *Wang et al., 2017*), eventually getting unloaded by XerD near the replication terminus (*Karaboja et al., 2021*). Bacterial genomes often have one or few closely positioned *parS* sites (separated by a few kb) (*Livny et al., 2007*). *Bacillus subtilis* (*Bsu*), however, harbors eight *parS* sites scattered over a much wider region of the genome (~0.75 Mb) (*Figure 1A*).

The two-sided DNA translocation by Smc-ScpAB is thought to have two main functions: (i) it organizes bacterial chromosomes by co-aligning chromosome arms as mentioned above and (ii) supports chromosome individualization presumably by localizing knots and precatenanes (i.e., DNA intertwinings) on the replicating chromosome, thus enabling DNA topoisomerases to completely untangle nascent sister DNA molecules efficiently (*Bürmann and Gruber, 2015*; *Orlandini et al., 2019*; *Racko et al., 2018*). This activity might be shared with condensin in eukaryotes (*Dyson et al., 2021*). The degree of defects in chromosome segregation caused by *smc* deletion is variable among species. In *B. subtilis*, chromosome segregation fails completely in *smc* mutants under nutrient-rich growth conditions but not when cells are grown with more limited nutrient availability (*Gruber et al., 2014*; *Orlandini et al., 2019*; *Wang et al., 2014*). Deletion of *parB* or removal of *parS* sites eliminate chromosome arm alignment, but have only a mild impact on chromosome segregation, demonstrating that chromosome arm alignment is not required for efficient chromosome segregation (and for cell viability) (*Lee et al., 2003*; *Wang et al., 2015*) and implying that Smc-ScpAB can use non-*parS* sequences for loading in the absence of ParB/*parS*. For simplicity, we represent the translocating unit of Smc-ScpAB as a single ring; however, we note that other arrangements (such as a double ring) are conceivable.

SMC complexes share a characteristic elongated architecture: a globular head and a hinge domain are connected by a long intramolecular antiparallel coiled coil 'arm' (*Figure 1B*). The functioning of the complex is restricted to discreet lengths of the coiled coil, the same periodicity of which is observed across diverse species and types of SMC proteins (*Bürmann et al., 2017*). Two SMC proteins dimerize at the hinge and are bridged at the head domains by a kleisin subunit. This generates annular tripartite SMC-kleisin assemblies that entrap chromosomal DNA double helices (*Gligoris et al., 2014*; *Wilhelm et al., 2015*). The kite subunit (ScpB in *B. subtilis*) also forms dimers that associate with the central region of kleisin (ScpA in *B. subtilis*) (*Bürmann et al., 2013*).

To support the nearly complete alignment of chromosome arms, Smc complexes must keep translocating on the same DNA molecule (i.e., remain in cis) and in the same direction for extended periods of time (estimated to be in the range of 40 min in *B. subtilis*). This processivity is thought to rely on the stable entrapment of one or more DNA double helices by the SMC complex guaranteeing lengthy periods of time between association and dissociation events ('chromosome residency') (*Gligoris et al., 2014*; *Wilhelm et al., 2015*). The extended nature of the coiled coils would nevertheless permit the SMC complex to overcome relatively big obstacles (~30 nm) without dissociating from DNA. How DNA entrapment might be compatible with the bypass of even larger obstacles on the chromosome remains unclear (*Brandão et al., 2019*). Moreover, Smc complexes loaded simultaneously at different *parS* sites (*Figure 1A*) will translocate towards each other and eventually collide. Dedicated mechanisms (such as Smc traversal proposed for purified yeast condensin) might be necessary to resolve such encounters. On the other hand, a translocation mechanism not involving DNA entrapment by the SMC complex would readily facilitate bypassing of obstacles, but it is unclear how directionality of translocation and chromosome association might be maintained in this case.

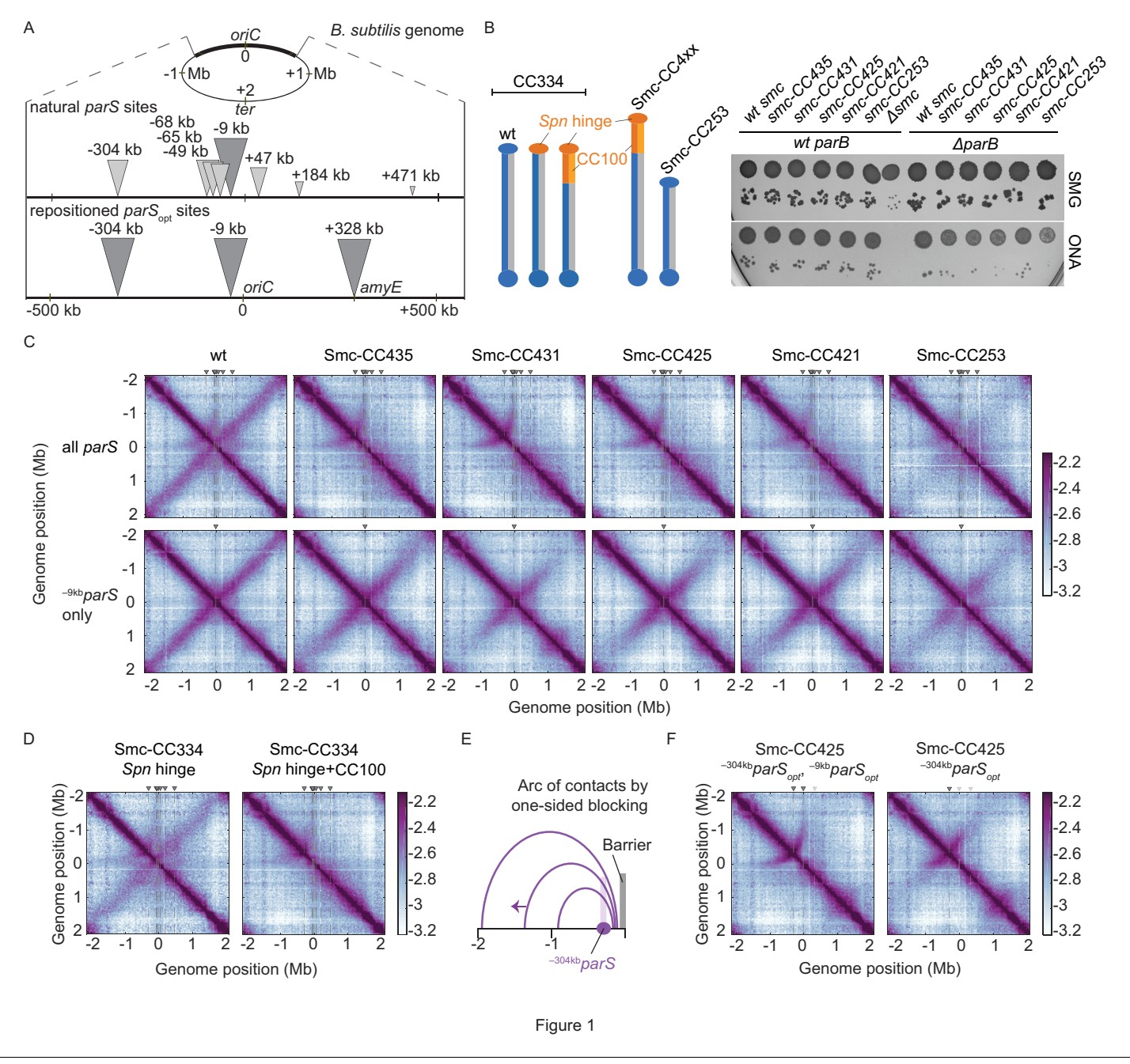

**Figure 1.** Arm-modified Smc proteins fail to align chromosome arms unless most *parS* sites are removed. (**A**) Upper panel: scheme depicting the natural distribution of *parS* sites on the *B. subtilis* genome. Triangles indicate positions of *parS* sites, size of which is scaled according to ParB occupancy judged by ChIP-seq (*Minnen et al., 2016*). Lower panel: scheme depicting engineered *parS* distribution generated in this study. *parS* sites were either eliminated by mutation or substituted for the *parS*$_{opt}$ sequence (i.e., the sequence of $^{-9kb}$*parS*) as needed. For some experiments, an additional site ($^{+328kb}$*parS*$_{opt}$) was inserted into the *amyE* locus. (**B**) Left panel: schemes of Smc coiled coil variants investigated in this study: wild-type (CC334), elongated (CC4xx), and shortened (CC253). *Spn* hinge+CC100, *Streptococcus pneumoniae* hinge domain, and 100 amino acids hinge-proximal coiled coil (in orange colors). The coiled coil was shortened or elongated starting from a chimeric protein having the *B. subtilis* Smc hinge domain replaced by the *S. pneumoniae* (*Spn*) Smc hinge domain including an ~100 aa (amino acid) segment of the adjacent coiled coil. Right panel: spotting assay of strains with altered Smc coiled coil in wild-type or sensitized background (Δ*parB*). $9^{-2}$ and $9^{-5}$ dilutions were spotted on nutrient-poor (SMG) or nutrient-rich medium (ONA) and imaged after 36 hr and 15 hr, respectively. Note that in the absence of ParB the ParABS system is non-functional and Smc loading is inefficient and untargeted, together putting a strain on chromosome segregation (*Minnen et al., 2016*; *Wilhelm et al., 2015*). The expression levels of some of these constructs (CC435, CC253) were previously shown to be close to the levels of the wild-type protein by immunoblotting (*Bürmann et al., 2017*). The levels of Smc-CC425 are evaluated in *Figure 1—figure supplement 1A*. (**C**) Normalized 3C-seq contact

*Figure 1 continued on next page*

*Figure 1 continued*

maps obtained from exponentially growing cultures. Top row: strains with wild-type *parS* sites. Bottom row: strains with a single $^{-9kb}parS_{opt}$ (*par-S359*) site. All 3C-seq maps presented in this study are split into 10 kb bins and have the replication origin placed in the middle. The interaction score is in $\log_{10}$ scale, the darker the color, the more interactions between given loci (see Materials and methods). (D) Normalized 3C-seq contact maps obtained from exponentially growing cultures carrying all the wild-type *parS* sites and wild-type length Smc (Smc-CC344) with either only hinge replaced by the *S. pneumoniae* sequence (*Spn* hinge, left panel) or the hinge together with 100 amino acids of hinge-proximal coiled coil replaced (*Spn* hinge + CC100, right panel). (E) Scheme for asymmetric loop extrusion starting at $^{-304kb}parS$ (*parS-334*) due to blockage of translocation towards the replication origin by head-on encounters (with other Smc complexes or RNA polymerase) generating an arc of contacts in the 3C-seq maps. (F) Normalized 3C-seq contact maps of elongated Smc (Smc-CC425) carrying *parS*<sub>opt</sub> sites at −304 kb and −9 kb (left panel) or *parS*<sub>opt</sub> site at −304 kb only (right panel). Triangles above the contact map point to positions of *parS* sites (dark gray triangles indicate active parS sites, light gray triangles for reference are *parS* sites absent in the given experiment).

The online version of this article includes the following figure supplement(s) for figure 1:

**Figure supplement 1.** Characterization of Smc variants, #1.
**Figure supplement 2.** Characterization of Smc variants, #2.
**Figure supplement 3.** Recruitment efficiency of various *parS* sequences.

Here, we studied the effect of Smc, ParB, and *parS* alterations on chromosome organization to explore how Smc-ScpAB load and translocate on a chromosome with multiple loading sites. Based on our results, we propose that Smc complexes rarely meet on the chromosome under physiological conditions. We argue that multiple parameters are fine-tuned to avoid Smc-Smc collisions in the first place. Few Smc complexes are available for loading because most Smc complexes are associated with the chromosome arms for an extended period of time. Artificially increasing the rate of encounters by mildly elevating the levels of Smc complexes in vivo, or by altering the efficiency of Smc loading, entrapment and/or translocation, leads to obvious perturbations in chromosome architecture, presumably due to unresolved Smc-Smc collisions. Also, the genomic clustering of strong *parS* sites seems to play a vital role in avoiding Smc collisions in *B. subtilis*. Although we cannot exclude the possibility of dedicated mechanism for the resolution of collisions per se, we suggest that an optimized Smc flux helps to eschew such events, presumably to avoid complications emerging from any attempted resolution reaction.

## Results

### Arm-modified Smc proteins fail to juxtapose chromosome arms

We previously isolated chimeric Smc proteins with elongated and shortened coiled coils that can functionally substitute for the *B. subtilis* Smc (**Bürmann et al., 2017**). From a collection of 20 constructs, we here identified several elongated Smc proteins, including Smc-CC425 (with a 425 aa coiled coil as compared to the 334 aa in wild-type Smc), which supported normal growth on nutrient-rich medium even in a sensitized background (ΔparB) (**Figure 1B**, **Figure 1—figure supplement 2D**). A selected Smc variant was tagged with HaloTag ('HT') to evaluate its expression levels. In-gel fluorescence detection showed comparable cellular levels for Smc-HT and Smc-CC425-HT proteins (**Figure 1—figure supplement 1A**). Chemical cross-linking of cysteine variants of Smc-CC425 moreover indicated that it assembles holo-complexes efficiently, and co-isolation of cross-linked circular species with intact chromosomal DNA implied a only slightly reduced fraction of chromosomally loaded Smc-CC425 (**Figure 1—figure supplement 1C**; **Vazquez Nunez et al., 2019**).

We hypothesized that the coiled coil length may influence DNA translocation, particularly when Smc complexes meet and overcome obstacles on the DNA track. To address this, we performed 3C-seq analysis on cells grown in nutrient-poor medium (SMG) at 37°C, supporting growth with a generation time of ~60 min. Encounters between translocating Smc complexes and the replication fork are expected to be rare under these conditions as replication initiates only about every 60 min (**Gruber et al., 2014**). We found that Smc-CC425 and the other elongated variants failed to support normal chromosome organization (**Figure 1C**). As revealed by the absence of a secondary diagonal, the co-alignment of chromosome arms was strongly compromised. An arc of contacts on the left arm of the chromosome however was observed in wild-type and mutant 3C-seq maps (see below). A control chimeric protein with wild-type arm length (Smc-CC334 *Spn* hinge+CC100) showed similar growth behavior and 3C-seq maps as the resized variants (**Figure 1D**, **Figure 1—figure supplement**

*1E*). With a shortened Smc protein, Smc-CC253, only residual levels of inter-arm contacts were noticeable, extending only few hundred kb from the replication origin (*Figure 1C*, *Figure 1—figure supplement 1D*). Another control chimeric protein with wild-type arm length and only the hinge domain replaced by the corresponding *Streptococcus pneumoniae* sequences (SmC-CC334 Spn hinge) was only moderately affected for chromosome organization (*Figure 1D*), implying that Smc arm modifications accounted for large parts of the defects in chromosome organization. We conclude that in contrast to wild-type Smc engineered Smc variants are unable to properly co-align the two chromosome arms despite supporting growth, and presumably chromosome segregation, apparently normally.

## Removal of all but one *parS* sites rescues chromosome folding by arm-modified Smc

To reveal the cause of the defect in chromosome arm alignment, we sought to characterize the loading and translocation of the modified Smc complexes on the bacterial chromosome. We started by generating strains in which seven *parS* sites were inactivated by mutations, with $^{-9kb}parS$ (*parS-359*) remaining the only *parS* site on the chromosome (along with the weak $^{+1058kb}parS$ site; *parS-90*; see below). As expected from published work (*Marbouty et al., 2015*; *Umbarger et al., 2011*; *Wang et al., 2015*), wild-type Smc efficiently aligned chromosome arms from a single strong *parS* site (*Figure 1C*). Remarkably, all four Smc proteins with an extended Smc arm displayed clearly increased levels of inter-arm contacts (*Figure 1C*). Near the replication origin, chromosome arm alignment was comparable to wild type, while the inter-arm contacts were less frequent (or absent) further away from the replication origin with all modified Smc constructs (*Figure 1C*; for quantification, see *Figure 1—figure supplement 2A*). The shortened Smc variant (CC253) also displayed more inter-arm contacts when the seven *parS* sites were mutated (*Figure 1C*). Thus, the removal of seven *parS* sites improved—rather than hindered—chromosome arm alignment by modified Smc proteins.

The arc of contacts detected on the left arm of the chromosome was lost in all strains harboring only the $^{-9kb}parS$ site (*Figure 1C*; *Marbouty et al., 2015*). It was also lost when only $^{-304kb}parS$ site (*parS-334*) was mutated but not when its sequence was substituted for the $^{-9kb}parS$ sequence (*Figure 1—figure supplement 2B, C*). Of note, the $^{-304kb}parS$ site is unique, in being relatively strong as well as distantly located from other strong *parS* sites (*Figure 1A*). DNA loop extrusion starting from this site is asymmetric, presumably due to a high likelihood of clashes with other Smc complexes and RNA polymerase (see scheme in *Figure 1E* and see below).

To test the impact of *parS* distribution in a more controlled way, we created strains with two distantly positioned *parS* sites (*Figure 1A*, lower panel). Since *parS* sites accumulate varying levels of ParB protein (*Graham et al., 2014*; *Minnen et al., 2016*), we first identified the *parS* sequences giving highest chromosomal recruitment of ParB and Smc when inserted at the *amyE* locus (+328 kb) in otherwise wild-type cells. The sequence of the $^{-9kb}parS$ site outperformed four other natural *parS* sequences and behaved equally well as an engineered consensus sequence at the ectopic location as judged by ChIP-qPCR (*Figure 1—figure supplement 3*). We thus used the strong $^{-9kb}parS$ sequence (denoted as $parS_{opt}$) in subsequent experiments. When two $parS_{opt}$ sites ($^{-9kb}parS_{opt}$ and $^{-304kb}parS_{opt}$) were combined on the same chromosome, chromosome arm alignment by Smc-CC425 became inefficient, producing a contact map similar to the one obtained with all *parS* sites present (*Figure 1F*).

RNA polymerase is a known impediment for Smc DNA translocation (*Brandão et al., 2019*). To dissect the contribution of Smc-Smc and Smc-RNA polymerase encounters in the hindrance of chromosome arm alignment, we treated exponentially growing cells for 15 min with the RNA polymerase inhibitor rifampicin ('rif') (at 25 ng/ml) (*Wang et al., 2017*). As seen before, the obtained maps became significantly noisier upon rif treatment (*Figure 1—figure supplement 2E*). The arc originating from $^{-304kb}parS$ in strains carrying Smc-CC425 was less pronounced in the presence of rif, indicating a partial relief of constraints on Smc translocation (*Figure 1—figure supplement 2E*). This indicates that transcription contributes to the defects observed with Smc-CC425 in the presence of multiple *parS* sites.

Altogether, we conclude that the presence of two or more *parS* sites hampers chromosome organization by Smc-CC425, conceivably because Smc-CC425 protein is more prone to collisions than wild-type Smc or less efficient in resolving them.

## An arm-modified Smc protein over-accumulates in the replication origin region

Wild-type Smc-ScpAB displays highest enrichment on the chromosome in the replication origin region with long and shallow gradients of enrichment along both chromosome arms (*Gruber and Errington, 2009*; *Minnen et al., 2016*), presumably generated by loading at *parS*, by translocation towards the replication terminus (*ter*) with some Smc being spontaneously unloaded from chromosome arms and the remaining fraction of Smc being unloaded by XerD near *ter* (*Karaboja et al., 2021*). Removal of seven *parS* sites had only a minor impact on the distribution of wild-type Smc-ScpAB as judged by chromatin immunoprecipitation coupled to deep sequencing (ChIP-seq) using α-ScpB serum (*Figure 2A, B*, left panels). The chromosomal distribution of Smc-CC425 was markedly different (*Figure 2A*). It showed hyper-enrichment near the replication origin and poor distribution towards the chromosome arms. Remarkably, the removal of seven *parS* sites substantially reduced the hyper-enrichment near the origin and increased the otherwise relatively low signal on the chromosome arms (*Figure 2A, B*, right panels, *Figure 2—figure supplement 1A*). The hyper-enrichment of Smc in the replication origin region thus correlated with poor chromosome arm alignment (*Figure 1C*). These results suggest that in the presence of multiple *parS* sites the modified Smc coiled coil either impedes Smc translocation provoking frequent collisions and unproductive loop formation or increases the rate of unloading and subsequent reloading events in the origin region. Both hypotheses could equally well explain the hyper-enrichment of Smc-CC425 in that region.

We next synchronized chromosomal loading of Smc and Smc-CC425 at a single *parS* site ($^{-9kb}$*parS*$^{opt}$) in a population of cells by depleting and repleting ParB protein. These experiments were performed at 30°C to allow sufficient ParB expression from a theophylline riboswitch-regulated *parB* construct. Smc and Smc-CC425 complexes were both found enriched in an ~700 kb region centered on the replication origin after 20 min of ParB induction by ChIP-seq analysis using α-ScpB serum (*Figure 2C*, *Figure 2—figure supplement 1B*). For wild-type Smc, the enriched region increased in size over time, inferring a constant DNA translocation rate of roughly 500 bp/s at 30°C. Notably, the high enrichment near *parS* disappeared at the later time points as Smc-ScpAB became more broadly distributed on the chromosome. The region of Smc-ScpAB enrichment also broadened in Smc-CC425 during the later time intervals, albeit with an apparently reducing rate. In addition, the origin region remained highly enriched in ScpB also at the later time points. Using 3C-seq, we observed that Smc-CC425 was able to align chromosome arms in this experimental system, yet the alignment did not extend all the way to the *ter* region (*Figure 2D*). Moreover, the onset of chromosome alignment as well as the rate of progress appeared somewhat reduced when compared to wild-type Smc. Determining a translocation rate for Smc-CC425 from the ChIP-Seq and 3C-Seq data turned out to be difficult because of the lack of a clear moving front particularly at the later time points. Moreover, the translocation rates appeared to reduce at later timer points, possibly due to increased spontaneous unloading of Smc-CC425. Regardless, these experiments demonstrate that Smc-CC425 efficiently accumulated in the replication origin region, but the translocation to distal loci on the chromosome arms was hampered.

A simple explanation for the hyper-accumulation of Smc-CC425 in the replication origin region (in the presence of a single *parS* site) is an increase in spontaneous unloading of translocating Smc. With shorter periods of time spent translocating along the chromosome arms, the cytoplasmic pool of Smc increases and as a consequence so does the flux of loading, which—possibly together with a reduced translocation rate—leads to artificially increased enrichment near the *parS* site(s). This in turn is expected to elevate the probability of Smc-Smc collisions on the chromosome when loading occurs at multiple *parS* sites but not when restricted to a single *parS* site. Such collisions would further exacerbate the Smc hyper-enrichment by hindering Smc translocation away from *parS* sites (*Figure 2A*, right panel). A reduced chromosome residency time and a reduced translocation rate may thus explain all phenotypic consequences of the Smc arm-modifications. Whether Smc-CC425 has a problem in resolving collisions remains to be established (see Discussion).

## Wild-type Smc protein generates overlapping chromosome folding patterns

We next wondered how wild-type Smc proteins co-align chromosome arms when starting DNA loop extrusion at multiple *parS* sites. Wild-type Smc displayed relatively low enrichment in the replication

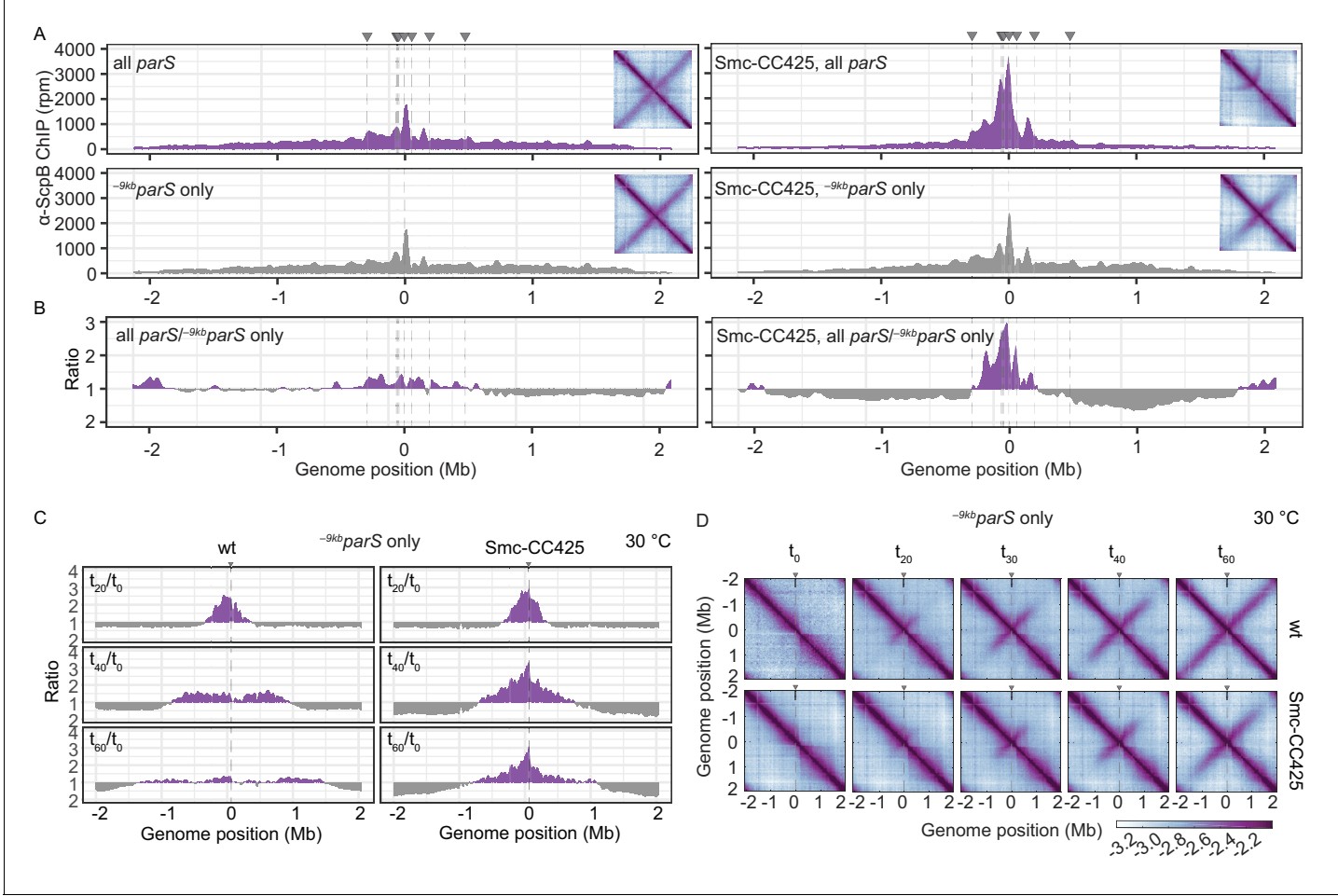

**Figure 2.** Modified Smc proteins hyper-accumulate in the replication origin region. (**A**) Read count distribution for chromatin immunoprecipitation coupled to deep sequencing (ChIP-seq) using α-ScpB serum. Left panel: strains carrying wild-type Smc with wild-type *parS* sites (top) or single $^{-9kb}parS_{opt}$ (*parS-359*) site (bottom). Removal of *parS* sites results in a slightly reduced enrichment in the origin region and in turn modestly increased signal mainly on the right arm of the chromosome (supposedly due to the presence of the weak $^{+1058kb}parS$ site; *parS-90*). Right panel: strains carrying Smc with elongated coiled coil (Smc-CC425) with wild-type *parS* sites (top) or single $^{-9kb}parS_{opt}$ (*parS-359*) site (bottom). Insets depict corresponding 3C-seq contact maps. All ChIP-seq profiles presented in this study are divided into 1 kb bins and have the replication origin placed in the middle. Dashed lines indicate the position of *parS* sites. (**B**) Ratio plots of ChIP-seq read counts for wild-type and elongated Smc (Smc-CC425) shown in (**A**). For each bin, normalized read counts for single $^{-9kb}parS_{opt}$ were compared with respective wild-type *parS* values. If the mutant/wild-type ratio was > 1, it was plotted above the genome position axis (in violet colors). If the mutant/wild-type ratio was < 1, the inverse ratio was plotted below the axis (in gray colors). (**C**) ChIP-seq time-course experiments using α-ScpB serum for strains carrying wild-type (left panel) or elongated Smc (Smc-CC425, right panel). These strains harbor a single loading site, $^{-9kb}parS_{opt}$ (*parS-359*), and a theophylline-inducible *parB* gene. Ratios plots of read counts for a given time point ($t_x$) versus $t_0$ are shown. For each bin, normalized read counts were compared with respective $t_0$ value and the higher value was divided by the lower. If the ratio $t_x/t_0$ was > 1, it was plotted above the genome position axis (in violet colors). If the ratio $t_0/t_x$ was > 1, the inverse ratio was plotted below the axis (in gray colors). (**D**) Normalized 3C-seq contact maps for the time course experiments with strains carrying wild-type (top panel) or elongated Smc (Smc-CC425, bottom panel), corresponding to (**C**).

The online version of this article includes the following figure supplement(s) for figure 2:

**Figure supplement 1.** Enrichment of Smc and Smc-CC425 in the replication origin region.

origin region even when all natural *parS* sites were present (***Figure 2A***). To understand how collisions between translocating Smc complexes are avoided or resolved, we next aimed to increase the incidence of collisions by positioning two $parS_{opt}$ sequences at selected sites in varying genomic distances and performed 3C-seq analysis.

As expected, control strains with a unique $parS_{opt}$ sequence at positions −9 kb, at −304 kb, or at +328 kb (at *amyE*) demonstrated extensive alignment of the respective flanking regions (***Figure 3A***; ***Wang et al., 2015***; ***Wang et al., 2017***). The $^{+328kb}parS_{opt}$ site, and to a lesser extent the $^-$

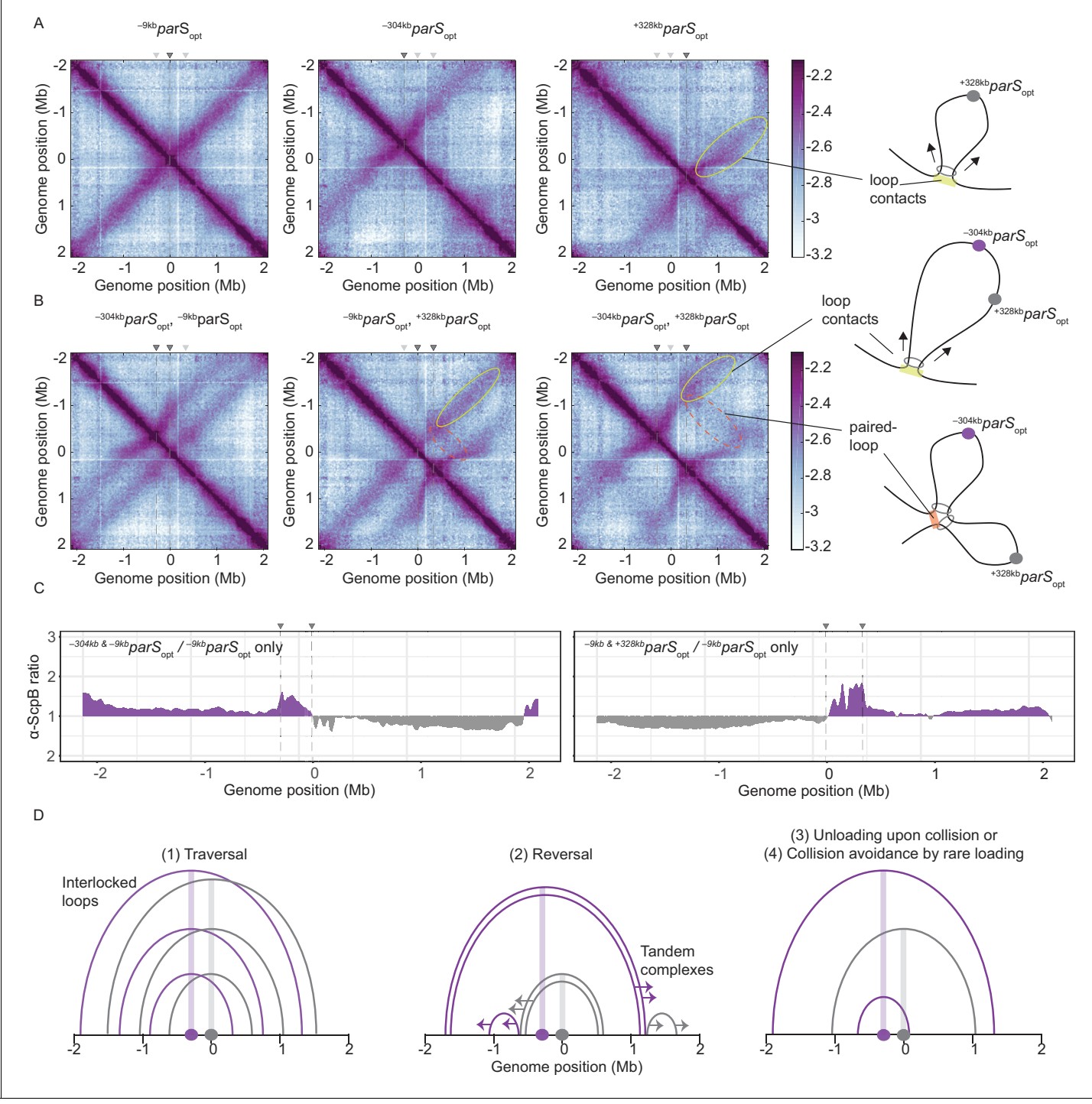

**Figure 3.** Overlapping chromosome arm alignment patterns for wild-type Smc. (**A**) Normalized 3C-seq contact maps for strains with a single *parS*<sub>opt</sub> site at −9 kb, −304 kb, or +328 kb. Dark gray triangles above the contact maps indicate the presence of active *parS* sites. Light gray triangles for reference are *parS* sites absent in the given experiment. Schemes depict a 'loop contact' that emerges by bidirectional translocation of a Smc unit from a single loading site (yellow), here *+328kb parS*<sub>opt</sub>. (**B**) Normalized 3C-seq contact maps for strains with two *parS*<sub>opt</sub> sites spaced by ~300 kb (left and middle) or ~600 kb (right). Schemes interpreting interactions in the contact maps: loop contacts (in yellow colors) and 'paired-loop contacts' that we presume to emerge by collision of two convergently translocating Smc units loaded at opposite *parS* sites (in orange colors). (**C**) Ratio plots for ChIP-seq read counts for a strain with two *parS* sites (left panel: *−304kb parS*<sub>opt</sub> and *−9kb parS*<sub>opt</sub>; right panel: *−9kb parS*<sub>opt</sub> and *+328kb parS*<sub>opt</sub>) and a control strain with a single *parS* site (*−9kb parS*<sub>opt</sub>). Representation as in *Figure 2B*. (**D**) Schemes depicting possible scenarios for long-distance contacts emerging by

*Figure 3 continued on next page*

*Figure 3 continued*

bidirectional Smc translocation with collision avoidance and collision resolution: Smc traversal (1), reversal (2), unloading upon collision (3), or low Smc flux (4).

The online version of this article includes the following figure supplement(s) for figure 3:

**Figure supplement 1.** Wild-type Smc protein generates overlapping chromosome folding patterns.

$^{304kb}parS_{opt}$ site, resulted in asymmetric alignment of the flanking DNA probably due to the presence of clusters of highly transcribed genes (including rDNA operons) in head-on orientation (*Brandão et al., 2019*). Of note, the frequency of contacts reaching beyond the replication origin is notably reduced with $^{-304kb}parS_{opt}$ or $^{+328kb}parS_{opt}$, implying that the origin region acts as a semi-permissive barrier for Smc translocation (or a Smc unloading site), as previously noted (*Minnen et al., 2016*). Note that in the map of Smc-CC425 with a single $^{-304kb}parS_{opt}$ site, only faint signals reaching beyond the origin at the secondary diagonal became visible (*Figure 1F*), being consistent with the replication origin region being a translocation barrier.

More importantly, when two *parS* sites were combined on the chromosome, striking novel patterns of chromosome organization by wild-type Smc arose (*Figure 3B*). In all cases, parallel secondary diagonals emerging from the two *parS* sites were detected. The pattern observed with $^{-304kb}parS_{opt}$ and $^{-9kb}parS_{opt}$ can—to a large degree—be explained as a combination of contacts observed in strains with the corresponding single *parS* sites, however, with clearly reduced probability for contacts extending beyond the region demarcated by the *parS* sites. A small but noticeable fraction of Smc complexes however apparently managed to translocate towards and beyond the other *parS* site mostly unhindered (as indicated by the largely unaltered position of the secondary diagonals). Treatment with rif did not significantly alter the contact pattern (apart from generating noisier maps) (*Figure 3—figure supplement 1A*). The contact maps involving the $^{-304kb}parS_{opt}$ and $^{+328kb}parS_{opt}$ sites showed additional contacts likely representing paired loops originating from collided Smc complexes loaded at opposite *parS* sites (*Figure 3B*). The presence of such paired loop contacts was less clear for the other *parS* combinations possibly due to background signal and limited resolution of the 3C-seq maps. We conclude that wild-type Smc-ScpAB complexes rarely block one another when loaded from all natural *parS* sites (with the notable exception of $^{-304kb}parS$). When the distance between two strong *parS* sites was artificially increased, however, impacts arising from collisions and blockage became noticeable. The blockage of Smc translocation was also apparent from ChIP-seq analysis, which demonstrated hyper-enrichment of Smc between two *parS* sites ($^{-304kb}parS_{opt}$ and $^{-9kb}parS_{opt}$ or $^{-9kb}parS_{opt}$ and $^{+328kb}parS_{opt}$) when compared to the single *parS* control (*Figure 3C*, *Figure 3—figure supplement 1C, D*). The effects of collisions on chromosome organization and Smc distribution are thus subdued with wild-type Smc but readily detectable upon repositioning of *parS* sites.

To explain the relatively mild impact of collisions in wild-type cells, we envisaged the following scenarios (*Figure 3D*, *Figure 3—figure supplement 1B*): (1) the traversal of Smc complexes generating interlocking loops (*Brandão et al., 2020*; *Kim et al., 2020*) (2) the reversal of the translocation of one Smc complex by opposing complexes (*Kim et al., 2020*), (3) the unloading of one or both complexes upon collision, or (4) collision avoidance either by infrequent loading or (5) by mutually exclusive *parS* usage. The latter hypothesis seemed highly unlikely as all but one *parS* sites would have to remain inactive for extended periods of time. While all other scenarios seemed plausible and may contribute to the process of chromosome organization, one scenario, the avoidance of encounters by infrequent loading, provided an explanation for the defects observed with Smc-CC425 without making additional assumptions.

## Increasing the pool of Smc hampers chromosome organization

If Smc-CC425 indeed fails to juxtapose chromosome arms due to an increased flux in the replication origin region, collisions may be rare in wild-type cells because of a high chromosome residency time and a limited pool of soluble Smc complexes, resulting in a low flux of Smc onto the chromosome. If so, artificially increasing the flux of Smc should lead to defects in chromosome organization with multiple *parS* sites but not with a single *parS* site (as observed for Smc-CC425 under normal expression levels) (assuming that most Smc is loaded at *parS*). If Smc complexes, however, were to

efficiently traverse, reverse, or unload one another, then increased Smc levels would not result in defective translocation and chromosome organization.

To test this prediction, we first slightly increased the cellular level of all subunits of the Smc complex by inserting an additional copy of the *smc* gene and of the *scpAB* operon under the control of their respective endogenous promoters into the genome. The increased levels of Smc-ScpAB did not noticeably affect cell growth (*Figure 4—figure supplement 1A*). Immunoblotting suggested a four- to fivefold increase in Smc and ScpB protein levels in the SMC^high strain when compared to wild type (*Figure 4A*). Next, we performed 3C-seq analysis. Chromosome arm co-alignment was strongly hampered—rather than improved—by the presence of extra Smc complexes in the cell (*Figure 4B*). A prominent arc was formed at the position of the $^{-304kb}$*parS* site, and the secondary diagonal originating in the origin region was weak and diffuse in the SMC^high strain. This defect was fully restored, however, by removal of seven *parS* sites (with the remaining strong site being either $^{-9kb}$*parS*$_{opt}$ or $^{-304kb}$*parS*$_{opt}$) (*Figure 4C*). Note that an additional feature (a minor secondary diagonal) present on the right arm of the chromosome likely originated from Smc loading at the weak $^{+1058kb}$*parS* site. The presence of two strong *parS* sites ($^{-9kb}$*parS*$_{opt}$ and $^{-304kb}$*parS*$_{opt}$) led to a new pattern of chromosome folding in the SMC^high strain. The alignment of DNA flanking the *parS* sites became highly asymmetric, presumably because Smc complexes loaded at opposite *parS* sites hinder each other (*Figure 4D*). Moreover, the contacts corresponding to paired loops became clearly visible (*Figure 4D*). Finally, contacts outside the *parS*-demarcated region were rare and spread out, and their center was shifted away from the *parS* sites. The former indicated that only a few Smc complexes loaded at one *parS* site managed to move beyond the other *parS* site. And if they did, they experienced a strongly reduced translocation rate when moving from *parS* site to the other, presumably due to encounters with and temporary (or partial) blockage by Smc complexes translocating in opposite orientation. Importantly, the hindrance observed with two or all *parS* sites being present was not relieved by treatment with rif, being consistent with the notion that Smc-Smc encounters rather than Smc-RNA polymerase encounters are mainly responsible for the impediment of translocation in SMC^high (*Figure 4—figure supplement 1B*).

If Smc-Smc collisions indeed hinder translocation of wild-type Smc, then extra Smc levels may lead to hyper-accumulation of Smc-ScpAB near the replication origin when multiple *parS* sites are present but not with a single *parS* site, as observed for the modified Smc at normal levels of expression (*Figure 2A*). To test this, we performed ChIP-seq with α-ParB and α-ScpB serum in SMC^high strains. The α-ParB ChIP-seq demonstrated that the localization of ParB to *parS* sites is, as expected, largely unaffected by the increased levels of Smc (*Figure 4E*; *Minnen et al., 2016*). The chromosomal distribution of ScpB was also largely unaffected in SMC^high cells harboring a single $^{-9kb}$parS$_{opt}$ site (*Figure 4—figure supplement 1C*). However, in the presence of two or multiple additional *parS* sites, the enrichment between the *parS* sites increased strongly (*Figure 4E, F*, *Figure 4—figure supplement 1C, D*). The changes in ScpB distribution upon *parS* site removal are remarkably similar in Smc-CC425 and SMC^high (compare *Figures 2B* and *4E*), supporting the notion that both modifications lead to more frequent blockage after collisions probably by the same mechanism: an increased flux of Smc in the vicinity of *parS* sites.

## Synchronized Smc loading favors Smc collisions

Finally, we synchronized the loading of Smc by induction of ParB with the idea that ParB repletion leads to a transiently elevated Smc flux (from a larger cytoplasmic pool of Smc) and thus increases the likelihood of encounters even with normal cellular levels of Smc-ScpAB. Here, we used a different inducible promoter, the IPTG-inducible $P_{spank}$ (*Wang et al., 2017*), which enabled us to grow cells at 37°C and compare the results more directly to the experiment with constitutively expressed ParB (*Figure 3B*). We found that the alignment of DNA starting from $^{-9kb}$*parS*$_{opt}$ site was indeed hampered when a second parS site, $^{-304kb}$*parS*$_{opt}$, was present (*Figure 4G*, *Figure 4—figure supplement 1E*), even more so than with continuous ParB expression (*Figure 3B*). However, we cannot rule out that this effect is mainly caused by competition between the two *parS* sites for Smc loading.

## Discussion

Establishing how SMC complexes manage to organize and orderly compact DNA in the crowded environment of a cell is a burning question in the field. SMC complexes translocate along an

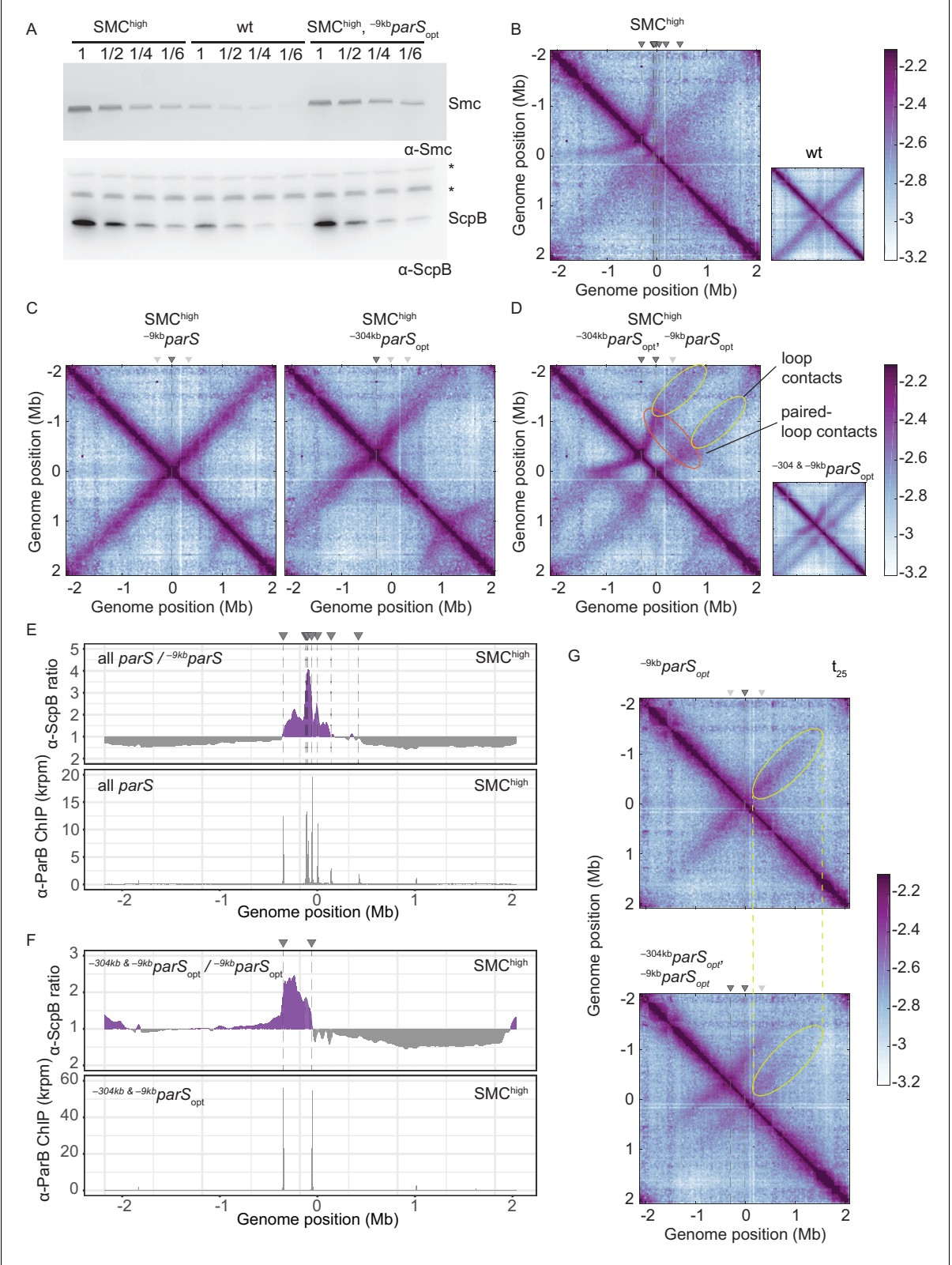

**Figure 4.** Increasing the cellular pool of Smc hampers chromosome organization. (A) Immunoblotting using α-Smc (top panel) and α-ScpB serum (bottom panel). SMC^high denotes strains with extra genes for Smc-ScpAB. Protein extracts of wild-type or SMC^high strains (harboring all *parS* sites or single *parS* site) were serially diluted with extracts from Δ*smc* or Δ*scpB* strains as indicated (see Materials and methods). * indicates unspecific bands generated by the α-ScpB serum. (B) Normalized 3C-seq contact map for SMC^high strain with all *parS* sites present. Inset shows 3C-seq contact map of a
*Figure 4 continued on next page*

*Figure 4 continued*

strain with wild-type protein levels (also displayed in *Figure 1C*) for direct comparison. (C) Normalized 3C-seq contact maps for SMC$^{high}$ strains with *parS*$_{opt}$ at −9 kb only or at −304 kb only. (D) Normalized 3C-seq contact map for SMC$^{high}$ strain with *parS*$_{opt}$ at positions: −9 kb and −304 kb. As in (B), with inset displaying respective control strain with normal Smc expression levels (also shown in *Figure 3B*). (E) Ratio plots for ChIP-seq read counts comparing SMC$^{high}$ strains with all *parS* sites and a single *parS* site ($^{−9kb}$*parS*$_{opt}$). Representation as in *Figure 2B* (top panel). Read count for α-ParB ChIP-seq in SMC$^{high}$ strain (bottom panel). (F) As in (E) involving a SMC$^{high}$ strain with two *parS* sites ($^{−304kb}$*parS*$_{opt}$ and $^{−9kb}$*parS*$_{opt}$) instead of all *parS* sites. (G) Normalized 3C-seq contact maps for time point t$_{25}$ after IPTG-induced ParB expression with a single *parS*$_{opt}$ site (top panel) or two *parS*$_{opt}$ sites (at −9 kb and −304 kb) (bottom panel). Ellipsoids (in yellow colors) mark the position of contacts stemming from loop extrusion originating at $^{−9kb}$*parS*$_{opt}$.

The online version of this article includes the following figure supplement(s) for figure 4:

**Figure supplement 1.** Synchronized loading of SMC hampers chromosome organization.

unusually flexible, congested and entangled translocation track, that is, the 'chromatinized' DNA double helix. The architecture of SMC complexes—one of a kind amongst the collection of molecular motors—is likely a reflection of a unique translocation mechanism. To support the folding of Mb-sized regions of the chromosome, Smc complexes need to keep translocating on the same DNA double helix from initial loading to unloading (a process lasting several tens of minutes in bacteria). Assuming a topological SMC-DNA association (*Gligoris et al., 2014*; *Wilhelm et al., 2015*), staying on the translocation track in cis is guaranteed as long as the SMC-kleisin ring remains closed, thus preventing the release of DNA.

During translocation, SMC complexes must frequently overcome obstacles on the DNA to translocate fast and far, and to globally organize the chromosome. RNA polymerase is a known obstacle for Smc-ScpAB in *B. subtilis*. It is highly abundant in the replication origin region due to the clustering of highly transcribed genes. Inhibition of RNA polymerase by the chemical inhibitor rif indeed partially relieved the impediment of DNA translocation by Smc. However, defects with SMC$^{high}$ in particular (*Figure 4—figure supplement 1B*) persisted in the presence of rif implying that other obstacles exist. Very large obstructions (>30 nm) could not be overcome while keeping DNA entrapped within the Smc ring but would need to be traversed by dissociating from the translocation track transiently. Such obstacles might include branched DNA structures and protein-mediated DNA junctions (i.e., crossings in the translocation track) as well as other SMC complexes located at the base of a DNA loop.

Traversal and bypassing are not risk-free strategies. When transiently disconnecting from DNA, the complex risks losing directional translocation by wrongly reconnecting with the same DNA double helix or even establishes an unwanted trans-DNA linkage by connecting with a different DNA double helix. Any straying onto the sister DNA molecule (going in trans) would not only defeat the purpose of DNA loop extrusion but actually actively hinder chromosome segregation. Here, we addressed the balance between avoiding and resolving Smc collisions.

## Avoiding Smc encounters

In our study, we show that impacts from collisions are barely noticeable in wild-type cells. Under physiological conditions, collisions between Smc-ScpAB complexes are kept at a tolerable level by a low cellular abundance of Smc-ScpAB ~30 Smc dimers per chromosome or less (*Wilhelm et al., 2015*), a high DNA translocation rate, an extended time of residency on the chromosome arms, and the preferential usage of a cluster of *parS* sites (spanning ~60 kb region of the genome: $^{−68kb}$*parS*, $^{−63kb}$*parS*, $^{−49kb}$*parS*, $^{−9kb}$*parS*). Thus, multiple parameters are fine-tuned to avoid Smc encounters, possibly rendering the resolution of Smc collisions unnecessary. Occasional loading of Smc at one of the more distal *parS* sites however would quite often lead to collisions, thus resulting in contact maps with an arc-shaped pattern, such as observed for $^{−304kb}$*parS* (the strongest of the distal *parS* sites) (*Figure 1C*; *Brandão et al., 2020*). When strong *parS* sites are artificially moved further away from each other, then impacts from collisions also become noticeable with wild-type Smc (*Figure 3*).

The obvious defects in chromosome organization observed with altered *parS* positioning, elevated Smc levels, or engineered Smc proteins, however, do not substantially impact bacterial growth, suggesting that chromosome segregation is efficiently supported even with Smc collisions (and without chromosome arm alignment being detectable by population-averaging 3C methodology). Colliding Smc complexes might thus efficiently promote DNA disentanglement locally (with

the help of DNA topoisomerases) but hamper the global chromosome folding process, possibly leading to large cell-to-cell variations in chromosome organization with likely knock-on effects on other cellular processes including nucleoid occlusion and cell division.

The presence of multiple types of SMC complexes acting on the same chromosome likely aggravates the issue of collisions. In *Pseudomonas aeruginosa*, the impact of collisions seems to be dealt with by a hierarchy amongst two endogenous SMC complexes and a coordination of SMC activity with chromosome replication (*Lioy et al., 2020*). Smc-ScpAB appears to limit the loop extrusion activity of MksBEF but not vice versa. When the heterologous *Escherichia coli* MukBEF complex was introduced in place of MksBEF, it blocked the activity of *P. aeruginosa* Smc-ScpAB but not the other way around. The hierarchy is possibly given by the relative abundance of these complexes and differing chromosome association dynamics (residency times).

## Resolving Smc encounters?

The parsimonious explanation for our observations—not requiring the involvement of dedicated and potentially hazardous molecular transactions—is that neither wild-type nor modified Smc proteins are able to traverse one another. A recent study describing Smc action in *B. subtilis* by simulations, however, suggested that Smc traversal is needed to accurately recapitulate the relative abundance of long-range contacts observed with natural and artificial arrangements of *parS* sites (similar to *Figure 3B*; *Brandão et al., 2020*). Briefly, the authors explained the patterns of DNA contact distribution on the basis of estimated or fitted values for Smc abundance (<40 per chromosome), Smc translocation rate (~1 kb/s), Smc-Smc blockage, Smc unloading (~0.0033 $s^{-1}$), and Smc traversal (~0.05 $s^{-1}$). While these simulations have apparently successfully predicted changes in contact patterns upon alterations in Smc abundance, we believe that they are not fully conclusive due to substantial uncertainties concerning the involved parameters and the possible existence of unexplored alternative scenarios (e.g., *Figure 3D*). Furthermore, direct observation of Smc traversal and of interlocking DNA loops (Z loops) on the bacterial chromosome is lacking. Also, a basic understanding of the necessary molecular transactions is elusive.

Nevertheless, the idea of Smc traversal is an intriguing proposition with potentially wide implications warranting serious consideration. A putative defect in traversal might contribute to the hyperaccumulation of Smc-CC425 in the replication origin region and the defective chromosome organization with multiple *parS* sites. Assuming the validity of Smc traversal, it is tempting to speculate—on the basis of the observations with Smc-CC425—that the nature and the integrity of the Smc hinge domain and the adjacent coiled coil are critical for the putative bypassing step. How Smc traversal might occur without the risk of establishing unwanted ('trans') DNA linkages yet is totally unclear.

## Multiple *parS* sites on the *B. subtilis* chromosome

The presence of multiple *parS* sites likely improves the robustness of the chromosome segregation process (*Böhm et al., 2020*). Most bacteria have clustered *parS* sites (within 5–40 kb region) and are sensitive to deleting or dispositioning them outside a tolerance region (*Böhm et al., 2020*; *Lagage et al., 2016*; *Minnen et al., 2011*; *Tran et al., 2017*). Severe consequences of manipulating *parS* distribution include a longer generation time (when *mksBEF* is missing) (*Lagage et al., 2016*), an increased number of anucleate cells (*Böhm et al., 2020*) and elongated cells (*Tran et al., 2017*). Some of these defects might be related to altered Smc function. In *Caulobacter crescentus*, Smc translocates only ~600 kb away from *parS* (*Tran et al., 2017*). The partial chromosome arm alignment in *C. crescentus* is reminiscent of the observation with modified Smc proteins in *B. subtilis*. It is tempting to speculate that a shorter chromosome residency time or a lower translocation rate of Smc-ScpAB is acceptable when *parS* sites are tightly clustered or when combined with a single *parS* site. Nevertheless, some bacterial genomes harbor multiple *parS* sites that are quite widely scattered on the genome. From the point of view of collision avoidance, this seems counterproductive. Intriguingly, the scattered *parS* distribution is restricted to few lineages on the phylogenetic tree of bacteria, including Bacilli (*Livny et al., 2007*). The scattering of *parS* sites likely serves a dedicated purpose in the lifestyles of these bacteria, in *B. subtilis* possibly during sporulation when large chunks of the genome (~1 Mb, i.e., about a quarter of the genome) are captured at the cell pole to promote entrapment of the chromosome in the pre-spore (*Wu and Errington, 1998*). This process might benefit from the condensation of the replication origin (possibly involving Smc collisions as

observed here) rather than an alignment of chromosome arms. Consistent with this notion, chromosome arm alignment is lost when DNA replication is artificially blocked (as naturally occurring during sporulation) and replaced by smaller loops formed at individual *parS* sites (*Marbouty et al., 2015*; *Wang et al., 2015*). A simple explanation for this altered pattern of chromosome folding during replication blockage (and possibly also during sporulation) would be an increased rate of Smc collisions due to an elevated (Smc) protein-to-DNA ratio.

Altogether, our results strongly suggest that the process of Smc DNA translocation is finely tuned to keep the probability of Smc encounters at a low level, presumably to enable extensive DNA loop extrusion without the need to resolve Smc collisions.

## Materials and methods

### *B. subtilis* strains and growth

*B. subtilis* 1A700 or PY79 isolate was used for experiments. Naturally competent *B. subtilis* was transformed via homologous recombination as described in *Diebold-Durand et al., 2019* and selected on SMG-agar plates with appropriate antibiotic selection. Transformants were next checked by PCR and, if required, Sanger sequencing. Genotypes of strains used in this study are listed in *Table 1*. More detailed information on how strains with mutated or wild-type *parS* sites were generated is provided in *Supplementary file 3*. Relevant, key plasmid maps are deposited in Mendeley Data DOI: 10.17632/kvjd6nj2bh.2.

For spotting assays, the cells were cultured in SMG medium at 37°C to stationary phase and $9^{-2}$ and $9^{-5}$-fold dilutions were spotted onto ONA (~16 hr incubation) or SMG (~36 hr incubation) agar plates.

### Immunoblotting

Cells were cultured in 150 ml of minimal media (SMG) at 37°C until mid-exponential phase ($OD_{600}$ = 0.022–0.025). Pellets were collected by filtration, washed, and resuspended in 1 ml PBSG (PBS supplemented with 0.1% glycerol). 1.25 $OD_{600}$ units of each sample were resuspended in 50 μl PBS containing 400 units of ReadyLyse lysosome (Epicentre), 12.5 units Benzonase (Sigma) and a protease-inhibitor cocktail (PIC, Sigma), and incubated for 30 min at 37°C. Next, 4× Loading Dye containing DTT (200 mM final) was added and samples were incubated for 5 min at 95°C. Protein extracts from tested strains were mixed with Δ*scpB* or Δ*smc* extracts as follows: tested strain only, 1:1 vol of tested strain with Δ, 1:4, 1:6. 5 μl of mixed protein extracts were run on Novex WedgeWell 4–12%, Tris-Glycine gels in 1× Laemmli buffer.

Proteins were transferred onto a PVDF membrane (Immobilon-P, Merck Millipore) using wet transfer. Membranes were blocked with 5% (w/v) milk powder in TBS with 0.05% Tween20. 1:2000 or 1:5000 dilutions of rabbit polyclonal sera against *B. subtilis* ScpB or Smc were used as primary antibodies for immunoblotting, respectively. The membrane was developed with HRP-coupled secondary antibodies and chemiluminescence (Amersham ECL Western Blotting Detection Reagent) and visualized on a FUSION FX7 (Vilber).

### Chromatin immunoprecipitation (ChIP)

ChIP samples were prepared as described previously (*Bürmann et al., 2017*). Briefly, cells were cultured in 200 ml of minimal media (SMG) at 37°C until mid-exponential phase ($OD_{600}$ = 0.022–0.030) and fixed with buffer F (50 mM Tris-HCl pH 7.4, 100 mM NaCl, 0.5 mM EGTA pH 8.0, 1 mM EDTA pH 8.0, 10% [w/v] formaldehyde) for 30 min at room temperature (RT) with occasional shaking. Cells were harvested by filtration and washed in PBS. Each sample was adjusted for 2 $OD_{600}$ units (2 ml at $OD_{600}$ = 1) and resuspended in TSEMS lysis buffer (50 mM Tris pH 7.4, 50 mM NaCl, 10 mM EDTA pH 8.0, 0.5 M sucrose and PIC [Sigma], 6 mg/ml lysozyme from chicken egg white [Sigma]). After 30 min of incubation at 37°C with vigorous shaking, protoplasts were washed again in 2 ml TSEMS, resuspended in 1 ml TSEMS, split into three aliquots, pelleted, and, after flash freezing, stored at −80°C until further use.

For time-course experiments, 1 l preculture was first grown until mid-exponential phase (OD = 0.022–0.030) and next appropriate culture volumes were added to fresh pre-warmed SMG so that at given time points 200 ml of culture at mid-exponential could be processed. The cultures were

**Table 1.** List of strains and genotypes used in the study.

| BSG | Genotype | Origin |
|-----|----------|--------|
| 1002 | 1A700, smc ftsY::ermB, trpC2 | The Gruber Laboratory |
| 1007 | 1A700, Δsmc ftsY::ermB, trpC2 | The Gruber Laboratory |
| 1018 | 1A700, smc(Streptococcus pneumoniae hinge) ftsY::ermB, trpC2 | The Gruber Laboratory |
| 1050 | 1A700, ΔparB::kanR, trpC2 | The Gruber Laboratory |
| 1471 | 1A700, smc(E1118Q) ftsY::ermB, ΔamyE::parS-359 + tetO qPCR primer seq::cat, trpC2 | This study |
| 1489 | 1A700, specR::scpA ΔscpB, trpC2 | The Gruber Laboratory |
| 1541 | 1A700, smc(E1118Q) ftsY::ermB, ΔamyE::parS-355 + tetO qPCR primer seq::cat, trpC2 | This study |
| 1542 | 1A700, smc(E1118Q) ftsY::ermB, ΔamyE::parS-354 + tetO qPCR primer seq::cat, trpC2 | This study |
| 1543 | 1A700, smc(E1118Q) ftsY::ermB, ΔamyE::parS-90 + tetO qPCR primer seq::cat, trpC2 | This study |
| 1544 | 1A700, smc(E1118Q) ftsY::ermB, ΔamyE::parS-optimal + tetO qPCR primer seq::cat, trpC2 | This study |
| 1711 | 1A700, specR::scpA ΔscpB, trpC2 | The Gruber Laboratory |
| 2090 | 1A700, smc(1-438, 487-684, 733-1186) ftsY::ermB, trpC2 | *Bürmann et al., 2017* |
| 2092 | 1A700, smc(1-399, 487-684, 772-1186) ftsY::ermB, trpC2 | *Bürmann et al., 2017* |
| 2093 | 1A700, smc(1-395, 487-684, 776-1186) ftsY::ermB, trpC2 | *Bürmann et al., 2017* |
| 2210 | 1A700, smc-HaloTag (C61V, C262A) ftsY::tetL ylqB, trpC2 | The Gruber Laboratory |
| 2352 | 1A700, smc(1-395, SpnSmc(398-768), 776-1186) ftsY::ermB, trpC2 | *Bürmann et al., 2017* |
| 2934 | PY79: Δ7-parS, parAB::kanR | This study |
| 3026 | PY79: Δ7-parS(parS359+), ΔparB::kanR, amyE::(PhbsB short 5'UTR-theo E+ -parB (mtparS))::CAT | This study |
| 3216 | PY79: Δ7-parS(parS359+), ΔparB::kanR, smc(1–476 SpnSmc(398-768) 695-1186) ftsY::ermB, amyE::(PhbsB short 5'UTR-theo E+ - parB (mtparS))::CAT | This study |
| 3425 | 1A700, smc(1-488), SpnSmc(398-768), smc(682-1186) ftsY::ermB, trpC2 | This study |
| 3426 | 1A700, smc(1-482), SpnSmc(398-768), smc(689-1186) ftsY::ermB, trpC2 | This study |
| 3427 | 1A700, smc(1-476), SpnSmc(398-768), smc(695-1186) ftsY::ermB, trpC2 | This study |
| 3428 | 1A700, smc(1-472), SpnSmc(398-768), smc(699-1186) ftsY::ermB, trpC2 | This study |
| 3429 | 1A700, ΔparB::kanR, smc(1-488), SpnSmc(398-768), smc(682-1186) ftsY::ermB, trpC2 | This study |
| 3430 | 1A700, ΔparB::kanR, smc(1-482), SpnSmc(398-768), smc(689-1186) ftsY::ermB, trpC2 | This study |
| 3431 | 1A700, ΔparB::kanR, smc(1-476), SpnSmc(398-768), smc(695-1186) ftsY::ermB, trpC2 | This study |
| 3432 | 1A700, ΔparB::kanR, smc(1-482), SpnSmc(398-768), smc(689-1186) ftsY::ermB, trpC2 | This study |
| 3636 | PY79: Δ7-parS, parAB::specR, smc(1-476 SpnSmc(398-768) 695-1186) ftsY::ermB | This study |
| 3674 | 1A700, Δ1-parS(mtparS334) | This study |
| 3770 | PY79: smc::tet, parAB::kanR | This study |
| 3785 | 1A700, ΔparB::kanR, smc(1-395, SpnSmc(398-768), 776-1186) ftsY::ermB, trpC2 | This study |
| 3786 | 1A700, ΔparB::kanR, smc(1-399, 487-684, 772-1186) ftsY::ermB, trpC2 | This study |
| 3787 | 1A700, ΔparB::kanR, smc(1-395, 487-684, 776-1186) ftsY::ermB, trpC2 | This study |
| 3790 | 1A700, Δ7-parS(parS359+), smc::specR | This study |
| 3791 | 1A700, mtparS334 to parS359 | This study |
| 3798 | 1A700, Δ1-parS(mtparS334), smc(1-476), SpnSmc(398-768), smc(695-1186) ftsY::ermB | This study |
| 3801 | 1A700, Δ7-parS, smc(1-488), SpnSmc(398-768), smc(682-1186) ftsY::ermB | This study |

*Table 1 continued on next page*

*Table 1 continued*

| BSG | Genotype | Origin |
|-----|----------|--------|
| 3802 | 1A700, Δ7-parS, smc(1-482), SpnSmc(398-768), smc(689-1186) ftsY::ermB | This study |
| 3803 | 1A700, Δ7-parS, smc(1-476), SpnSmc(398-768), smc(695-1186) ftsY::ermB | This study |
| 3804 | 1A700, Δ7-parS, smc(1-472), SpnSmc(398-768), smc(699-1186) ftsY::ermB | This study |
| 3805 | 1A700, Δ6-parS (mtparS334 to parS359), smc::ermB | This study |
| 3815 | 1A700, Δ8-parS, parAB(mtparS359)::kanR, smc::specR | This study |
| 3840 | 1A700, Δ7-parS(parS359+), ΔparB::kanR, smc::specR | This study |
| 3841 | 1A700, Δ7-parS(parS359+), ΔparB::kanR, smc(1-488), SpnSmc(398-768), smc(682-1186) ftsY::ermB | This study |
| 3842 | 1A700, Δ7-parS(parS359+), ΔparB::kanR, smc(1-482), SpnSmc(398-768), smc(689-1186) ftsY::ermB | This study |
| 3843 | 1A700, Δ7-parS(parS359+), ΔparB::kanR, smc(1-476), SpnSmc(398-768), smc(695-1186) ftsY::ermB | This study |
| 3844 | 1A700, Δ7-parS(parS359+), ΔparB::kanR, smc(1-472), SpnSmc(398-768), smc(699-1186) ftsY::ermB | This study |
| 3863 | PY79: smc(1-476), SpnSmc(398-768), smc(695-1186) ftsY::ermB, parAB::kanR | This study |
| 3878 | 1A700, Δ6-parS(parS359+, parS334->359+), smc(1-476), SpnSmc(398-768), smc(695-1186) ftsY::tetR | This study |
| 3879 | 1A700, Δ7-parS(parS334->359+), parAB::kanR, smc::ermB | This study |
| 3882 | 1A700, Δ7-parS(parS334->359+), parAB::kanR, smc(1-476), SpnSmc(398-768), smc(695-1186) ftsY::specR | This study |
| 3932 | 1A700, Δ7-parS(parS359+), smc(1-395, 487-684, 776-1186) ftsY::ermB | This study |
| 4083 | 1A700, Δ6-parS(parS359+), smc::specR, ΔamyE::parS-359 + tetO qPCR primer seq::cat | This study |
| 4090 | 1A700, Δ7-parS, parAB::kanR, smc::specR, ΔamyE::parS-359 + tetO qPCR primer seq::cat | This study |
| 4091 | 1A700, Δ6-parS(parS334->359+), parAB::kanR, smc::ermB, ΔamyE::parS-359 + tetO qPCR primer seq::cat | This study |
| 4100 | 1A700, ΔamyE::smc::CAT, qoxD-specR::scpAB-ywcE, trpC2 | This study |
| 4137 | 1A700, Δ6-parS (mtparS334 to parS359), smc::ermB, ΔamyE::smc::CAT, qoxD-specR::scpAB-ywcE | This study |
| 4143 | 1A700, Δ7-parS(parS359+), smc::ermB, ΔamyE::smc::CAT, qoxD-specR::scpAB-ywcE | This study |
| 4146 | 1A700, Δ7-parS(parS334->359+), parAB::kanR, smc::ermB, ΔamyE::smc::CAT, qoxD-specR::scpAB-ywcE | This study |
| 4152 | 1A700, Δ7-parS(parS359+), ΔparB::kanR, smc::specR, ΔamyE::(Pspank-optRBS-parB(mtparS)-lacI)::CAT | This study |
| 4427 | 1A700, Δ6-parS (mtparS334 to parS359), ΔparB::kanR, smc::specR, ΔamyE::(Pspank-optRBS-parB(mtparS)-lacI)::CAT | This study |
| 4798 | 1A700, smc(1-476), SpnSmc(398-768), smc(695-1186)-TEV-HaloTag ftsY::ermB, specR::scpA(E52C, H235C), trpC2 | This study |
| 4837 | 1A700, smc(S152C, R1032C)-TEV-HaloTag ftsY::ermB, specR::scpA(E52C, H235C), trpC2 | This study |
| 4838 | 1A700, smc(1-476)(S19C, S152C), SpnSmc(398-768), smc(695-1186)(R1032C)-TEV-HaloTag ftsY::ermB, specR::scpA(E52C, H235C), trpC2 | This study |
| 4867 | 1A700, smc(1-476)(S152C), SpnSmc(398-768), smc(695-1186)(R1032C)-TEV-HaloTag, specR::scpA(E52C, H235C), trpC2 | This study |
| 4869 | 1A700, smc(S19C, S152C, R1032C)-TEV-HaloTag ftsY::ermB, specR::scpA(E52C, H235C), trpC2 | This study |

induced with 2 mM theophylline ($P_{theo}$ promoter). Due to characteristics of the theophylline switch, the pre-culture as well as induction was performed at 30°C.

For ChIP-qPCR, each pellet was resuspended in 2 ml of buffer L (50 mM HEPES-KOH pH 7.5, 140 mM NaCl, 1 mM EDTA pH 8.0, 1% [v/v] Triton X-100, 0.1% [w/v] Na-deoxycholate, 0.1 mg/ml RNaseA and PIC [Sigma]) and transferred to 5 ml round-bottom tubes. Cell suspensions were sonicated 3 × 20 s on a Bandelin Sonoplus with a MS72 tip (90% pulse and 35% power output). Next, lysates were transferred into 2 ml tubes and centrifuged 10 min at 21,000 g at 4°C. 800 µl of supernatant was used for IP and 200 µl was kept as whole-cell extract (WCE).

For IP, first, antibody serum was incubated with Protein G coupled Dynabeads (Invitrogen) in 1:1 ratio for 2.5 hr at 4°C with rotation. Next, beads were washed in buffer L and 50 µl were aliquoted to each sample tube. Samples were incubated for 2 hr at 4°C with rotation, followed by a series of washes with buffer L, buffer L5 (buffer L containing 500 mM NaCl), buffer W (10 mM Tris-HCl pH 8.0, 250 mM LiCl, 0.5% [v/v] NP-40, 0.5% [w/v] sodium deoxycholate, 1 mM EDTA pH 8.0), and buffer TE (10 mM Tris-HCl pH 8.0, 1 mM EDTA pH 8.0). Finally, the beads were resuspended in 520 µl buffer TES (50 mM Tris-HCl pH 8.0, 10 mM EDTA pH 8.0, 1% [w/v] SDS). 300 µl of TES and 20 µl

of 10% SDS were also added to WCE. Both tubes were incubated O/N at 65°C with vigorous shaking to reverse formaldehyde crosslinks.

Phenol-chloroform extraction was performed to purify the decrosslinked DNA. Samples were transferred to screw cap 1.5 ml tubes and first mixed vigorously with 500 μl of phenol equilibrated with buffer (10 mM Tris-HCl pH 8.0, 1 mM EDTA). After centrifugation (10 min, RT, 13,000 rpm), 450 μl of the aqueous phase was transferred to a new screw cap tube and mixed with equal volume of chloroform, followed by centrifugation. 400 μl of aqueous phase was recovered for DNA precipitation with 2.5× volume of 100% ethanol, 0.1× volume of 3 M NaOAc, and 1.2 μl of GlycoBlue and incubated for 20 min at −20°C. Next, samples were centrifuged for 10 min at 20,000 g at RT and pellets obtained pellets were resuspended in 10 μl of EB (Qiagen) shaking at 55°C for 10 min and finally purified with a PCR purification kit, eluting in 50 μl EB.

For qPCR, 1:10 and 1:1000 dilutions in water of IP and WCE were prepared, respectively. Each 10 μl reaction was prepared in duplicate (5 μl Takyon SYBR MasterMix, 1 μl 3 μM primer pair, 4 μl of DNA) and run in Rotor-Gene Q machine (Qiagen). Primer sequences are listed in *Table 2*. Data was analyzed using PCR Miner server (http://ewindup.info) (*Zhao and Fernald, 2005*).

For IP of samples for ChIP-seq, the procedure was the same as for ChIP-qPCR, except for resuspending the pellets in 1 ml of buffer L and sonication in a Covaris E220 water bath sonicator for 5 min at 4°C, 100 W, 200 cycles, 10% load, and water level 5.

For deep sequencing, the DNA libraries were prepared by Genomic Facility at CIG, UNIL, Lausanne. Briefly, the DNA was fragmented by sonication (Covaris S2) to fragment sizes ranging from 220 to 250 bp. DNA libraries were prepared using the Ovation Ultralow Library Systems V2 Kit (NuGEN) including 15 cycles of PCR amplification. 5–10 million single-end sequence reads were obtained on a HiSeq2500 (Illumina) with 150 bp read length.

## Processing of ChIP-seq reads

Reads were mapped to *B. subtilis* genome NC_000964.3 (for 1A700) or NC_0022898 (for PY79) with bowtie2 using –very-sensitive-local mode. Subsequent data analysis was performed using Seqmonk (http://www.bioinformatics.babraham.ac.uk/projects/seqmonk/) and R. The bin size used is 1 kb. For the enrichment plots, the data was smoothened using Local Polynomial Regression Fitting (loess).

## Generation of chromosome conformation capture (3C) libraries

3C libraries were prepared as previously described (*Marbouty et al., 2015*). Minimal media (SMG) was used instead of LB. Briefly, cells were grown in 400 ml of SMG medium to exponential phase ($OD_{600}$ = 0.022–0.030) and fixed with fresh formaldehyde (3% final concentration) for 30 min at RT, followed by 30 min at 4°C, and quenched for 30 min with 0.25 M glycine at 4°C. Fixed cells were harvested by filtering, washed in fresh SMG, frozen in liquid nitrogen, and stored at −80°C until further use.

Samples for RNAP inhibition experiment were prepared as other 3C libraries with additional rifampicin treatment before harvesting. Exponentially growing cultures were split in 2 × 400 ml cultures (two technical replicates for each, treated sample and untreated control). One was treated with 25 ng/μl rifampicin for 15 min (duration and concentration of treatment as in *Wang et al., 2017*) and respective control sample was left to grow for 15 min without treatment.

**Table 2.** List of oligos used for the qPCR.

| Locus | Oligo name1 | Oligo sequence1 | Oligo name2 | Oligo sequence2 |
| --- | --- | --- | --- | --- |
| *parS354* | STG495 | ttgcagctaactgccatttg | STG496 | aaaactgaacaggggtcacg |
| *parS355* | STG493 | taattcatcatcgcgctcaa | STG494 | aatgccgattacgagtttgc |
| *parS359* | STG097 | aaaaagtgattgcggagcag | STG098 | agaaccgcatctttcacagg |
| *parS90* | STI587 | gccattgggcatcagtatg | STI588 | ataagcgacaccttgctcgt |
| *dnaA* | STG199 | gatcaatcggggaaagtgtg | STG200 | gtagggcctgtggatttgtg |
| *amyE* | STG220 | aatcgtaatctgggcgtgtc | STG221 | catcatcgctcatccatgtc |
| *ter* | STG099 | tccatatcctcgctcctacg | STG100 | attctgctgatgtgcaatgg |

For time-course experiments, 2 l preculture was first grown until mid-exponential phase (OD = 0.022–0.030) and next appropriate culture volumes were added to fresh pre-warmed SMG so that at given time points 2 × 200 ml of culture at mid-exponential could be collected (two technical replicates). The cultures were induced with 2 mM theophylline or 1 mM IPTG, depending on the promoter used, $P_{theo}$ or $P_{spank}$, respectively. Due to the characteristics of the theophylline switch, the pre-culture as well as induction was performed at 30℃.

Frozen pellets were resuspended in 600 µl 1× TE and incubated at RT for 20 min with 4 µl of Ready-lyze lysozyme (35 U/µl, Tebu Bio). Next, SDS was added to a final concentration of 0.5% and incubated at RT for 10 min. 50 µl of lysed cells were aliquoted to eight tubes containing 450 µl of digestion mix (1× NEB 1 buffer, 1% triton X-100, and 100 U HpaII enzyme [NEB]) and incubated at 37℃ for 3 hr with constant shaking. Digested DNA was collected by centrifugation, diluted into four tubes containing 8 ml of ligation mix (1× ligation buffer: 50 mM Tris-HCl, 10 mM MgCl$_2$, 10 mM DTT), 1 mM ATP, 0.1 mg/ml BSA, 125 U T4 DNA ligase 5 U/ml, and incubated at 16℃ for 4 hr. Ligation reaction was followed by O/N decrosslinking at 65℃ in the presence of 250 µg/ml proteinase K (Eurobio) and 5 mM EDTA.

DNA was precipitated with 1 vol of isopropanol and 0.1 vol of 3 M sodium acetate (pH 5.2, Sigma) at −80℃ for 1 hr. After centrifugation, the DNA pellet was resuspended in 1× TE at 30℃ for 20 min. Next, DNA was extracted once with 400 µl phenol-chloroform-isoamyl alcohol solution and precipitated with 1.5 vol cold 100% ethanol in the presence of 0.1 vol 3 M sodium acetate at −80℃ for 30 min. The pellet was collected and resuspended in 30 µl TE with RNaseA at 37℃ for 30 min. All tubes were pooled and the resulting 3C library was quantified on gel using ImageJ.

## Processing of libraries for Illumina sequencing

1 µg of 3C library was suspended in water (final volume 130 µl) and sonicated using Covaris S220 (following the manufacturer's recommendations to obtain 500 bp target size). Next, DNA was purified with Qiagen PCR purification kit, eluted in 40 µl EB, and quantified using NanoDrop. 1 µg of DNA was processed according to the manufacturer's instructions (Paired-End DNA sample Prep Kit, Illumina, PE-930-1001), except that DNA was ligated to custom-made adapters for 4 hr at RT, followed by inactivation step at 65℃ for 20 min. DNA was purified with 0.75× AMPure beads and 3 µl were used for 50 µl PCR reaction (12 cycles). Amplified libraries were purified on Qiagen columns and pair-end sequenced on an Illumina platform (HiSeq4000 or NextSeq).

## Processing of PE reads and generation of contact maps

Sequencing data was demultiplexed, adapters trimmed, and PCR duplicates removed using custom scripts. Next, data was processed as described at https://github.com/axelcournac/3C_tutorial. Briefly, bowtie2 in -very sensitive-local mode was used for mapping for each mate. After sorting and merging both mates, the reads of mapping quality >30 were filtered out and assigned to a restriction fragment. Uninformative events like recircularization on itself (loops), uncut fragments, and religations in original orientation were discarded (Cournac et al., 2012) and only pairs of reads corresponding to long-range interactions were used for generation of contact maps (between 5 and 8% of all reads). The bin size used is 10 kb. Next, contact maps were normalized through the sequential component normalization procedure (SCN, Cournac et al., 2012). Subsequent visualization was done using MATLAB (R2019b). To facilitate visualization of the contact matrices, first we applied to the SCN matrices the $\log_{10}$ and then a Gaussian filter (H = 1) to smooth the image. The scale bar next to the maps represents the contact frequencies in $\log_{10}$—the darker the color, the higher the frequency of contacts between given loci.

## Expression analysis of HaloTagged proteins

Cells were cultured at 37℃ in 200 ml SMG to exponential phase (OD$_{600}$ = 0.022–0.030) and harvested by filtration. Next, they were washed in cold PBS supplemented with 0.1% (v/v) glycerol ('PBSG') and split into aliquots of a biomass equivalent to 1.25 OD$_{600}$ units. Cells were centrifuged for 2 min at 10,000 g, resuspended in 40 µl PBSG containing 75 U/ml ReadyLyse Lysozyme, 750 U/ml Sm nuclease, 5 µM HaloTag TMR Substrate and protease inhibitor cocktail ('PIC') and incubated at 37℃ for 30 min to facilitate lysis. After lysis, 10 µl of 4× LDS-PAGE with DTT (200 mM final) buffer

was added, samples were incubated for 5 min at 95℃, and resolved by SDS-PAGE. Gels were imaged on an Amersham Typhoon scanner with Cy3 DIGE filter setup.

## Chromosome co-entrapment assay

Microbead entrapment followed the developments reported in *Vazquez Nunez et al., 2019*. Cells containing the Smc alleles with cysteines at the desired positions were inoculated in SMG medium to $OD_{600}$ = 0.004 and grown to mid-exponential phase ($OD_{600}$ = 0.02) at 37℃. Cells were mixed with ice for rapid cooling and harvested by filtration. A cell mass equivalent of $OD_{600}$ units of 3.75 was resuspended in 121 µl PBSG and incubated with a final concentration of 1 mM BMOE for 10 min. Reactions were quenched by the addition of β-mercaptoethanol to a final concentration of 32.6 mM. 45 µl of cross-linked cells were retained as 'Input' sample. 1 µl of PIC and 9 µl Dynabeads Protein G were added to 90 µl of the remaining cell fraction. Samples for entrapment were mixed with 100 µl 2% low-melt agarose at a temperature of 45℃ before being mixed rapidly with 700 µl mineral oil. Resulting Agarose microbeads were washed once in 1 ml RT PBSG by centrifugation at 10,000 rpm for 1 min. Beads were subsequently resuspended in 300 µl PBSG and mixed with EDTA pH 8 (1 mM final), 5 µl PIC, Halo TMR ligand (5 µM final), as well as ReadyLyse lysozyme to a final concentration of 40 U/µlL. Input samples were mixed with 5 µlL of a master mix containing 0.9 µl PBSG, 0.5 µl PIC, 2.5 µlL 1:100 Benzonase, 0.5 µl Halo TMR ligand, as well as 0.6 µlL of a 1:10 dilution of ReadyLyse lysozyme. Input as well as Entrapment samples were incubated for 25 min at 37˚. All subsequent steps were undertaken to protect from light as much as possible. Input samples were mixed with 50 µl 2× LDS loading dye. Entrapment samples were washed twice with 1 ml PBSG by centrifugation at 10,000 rpm, 1 min, RT. Microbeads were then washed three times in TES under gentle (500 rpm) shaking, first for 1 hr with two subsequent washing steps for 30 min each. Preparations were resuspended in 1 ml TES and incubated on a rolling incubator overnight at 4℃. Subsequently, the beads were washed twice with 1 ml PBS before being resuspended in 100 µl PBS. 5 µl Benzonase were added (750 U/ml final concentration), and samples were incubated at 37℃ under light shaking for 1 hr. To free the preparations from agarose, the samples were first heated to 70℃ for 1 min before incubating on ice for 5 min. Agarose was removed from the sample content by centrifugation, first with 21,000 rpm at 4℃ for 15 min, then with 14,000 rpm at RT for 5 min. Supernatant liquid was transferred to spin columns and centrifuged for 1 min, 10,000 rpm, ambient temperature. The resulting solution was brought to 1 ml total volume with water and mixed to final concentrations of 33 mg/ml BSA and 0.02% (w/v) deoxycholate before resting on ice for 30 min. Trichloroacetic acid was added to a final concentration of 8.8% preceding a 1 hr incubation on ice. Precipitated protein was spun down (21,000 rpm, 4℃, 15 min), resuspended in 10 µl 1× LDS loading dye and brought to neutral pH by µl-wise addition of 1 M Tris solution. 5 µl of input material and all of the 'eluate' samples were loaded on Tris-acetate gels and run at 35 mA. Gels were scanned using Amersham Typhoon Imager with the Cy5 filter settings.

## Acknowledgements

We thank Hugo Brandao, Leonid Mirny, and Xindan Wang for sharing unpublished data and comments on the manuscript. We are grateful to Frank Bürmann and all members of the Gruber laboratory for stimulating discussions and critical feedback. We thank Marc Garcia-Garcera and Björn Vessman for continuous support with R and Python and the Genomic Technologies Facility (GTF, UNIL, Lausanne) and Next Generation Sequencing Core Facility (CNRS, I2BC, Gif-sur-Yvette) for ChIP-seq library preparation and deep sequencing. This work was supported by funding from the European Research Council (Horizon 2020 ERC CoG 724482) (SG) and the Agence Nationale pour la Recherche (ANR-CE12-0013-01) and the Assocation pour la Recherche contre le Cancer (FB).

## Additional information

### Funding

| Funder | Grant reference number | Author |
|---|---|---|
| H2020 European Research Council | 724482 | Stephan Gruber |

| Agence Nationale de la Recherche | ANR-CE12-0013-01 | Frederic Boccard |
| Assocation pour la Recherche contre le Cancer | | Frederic Boccard |

The funders had no role in study design, data collection and interpretation, or the decision to submit the work for publication.

### Author contributions

Anna Anchimiuk, Conceptualization, Data curation, Formal analysis, Validation, Investigation, Visualization, Methodology, Writing - original draft, Writing - review and editing; Virginia S Lioy, Data curation, Software, Formal analysis, Visualization, Methodology; Florian Patrick Bock, Resources, Investigation; Anita Minnen, Data curation, Investigation; Frederic Boccard, Supervision, Funding acquisition, Methodology, Writing - review and editing; Stephan Gruber, Conceptualization, Formal analysis, Supervision, Funding acquisition, Visualization, Writing - original draft, Project administration, Writing - review and editing

### Author ORCIDs

Anna Anchimiuk https://orcid.org/0000-0002-2899-9413
Stephan Gruber https://orcid.org/0000-0002-0150-0395

### Decision letter and Author response

Decision letter https://doi.org/10.7554/eLife.65467.sa1
Author response https://doi.org/10.7554/eLife.65467.sa2

## Additional files

### Supplementary files

• Supplementary file 1. Information about replicates for each relevant figure panel.

• Supplementary file 2. List of strains sorted according to relevant figure panels and ordered in the way they are presented.

• Supplementary file 3. Details of strain construction. Key plasmid maps for generating the strains are deposited in Mendeley Data DOI: 10.17632/kvjd6nj2bh.2.

• Supplementary file 4. Table listing strains for which 3C-seq maps were generated with reference to the figure panels, genotype, and number of valid reads.

• Transparent reporting form

### Data availability

All deep sequencing data has been deposited to the NCBI GEO database and will be available at GEO Accession number: GSE163573 All other raw data will be made available via Mendeley Data https://doi.org/10.17632/kvjd6nj2bh.2.

The following datasets were generated:

| Author(s) | Year | Dataset title | Dataset URL | Database and Identifier |
|---|---|---|---|---|
| Anchimiuk A, Lioy VS, Boccard F, Gruber S | 2020 | GEO_Smc_collisions | https://www.ncbi.nlm.nih.gov/geo/query/acc.cgi?acc=GSE163573 | NCBI Gene Expression Omnibus, GSE163573 |
| Anchimiuk A, Gruber S | 2021 | Smc_collisions | https://data.mendeley.com/datasets/kvjd6nj2bh/2 | Mendeley Data, 10.17632/kvjd6nj2bh.2 |

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
