## [Decision Letter]

**Acceptance summary:**

This manuscript presents intriguing data to support the notion that *B. subtilis* cells have tuned a variety of parameters related to SMC loading and translocation to ensure that individual complexes do not collide. This is likely an important but poorly understood aspect of condensins/SMCs, and as such represents a valuable contribution to the field and should be of interest to a broad set of readers.

**Decision letter after peer review:**

Thank you for submitting your article "Fine-tuning of the Smc flux facilitates chromosome organization in *B. subtilis*" for consideration by *eLife*. Your article has been reviewed by 3 peer reviewers, and the evaluation has been overseen by a Reviewing Editor, Michael Laub, and Jessica Tyler as the Senior Editor. The reviewers have opted to remain anonymous.

Essential Revisions:

Although the reviewers were generally enthusiastic, there were some concerns raised. The full set of concerns/issues is appended below and the authors should respond to each item in a revised document. Immediately below is a summary of the 5 major issues – the first two include additional experiments that the reviewers collectively decided would be needed for a revision.

1) The authors need to include a quantification of SMC (and SMC variant) levels, including an assessment by chromatin fractionation of the quantity free in solution versus chromosome-associated. There are also several cases of quantification of Hi-C data that are needed.

2) Additional experiments are needed to help address the relative contributions of SMC-RNAP and SMC-SMC collisions. In particular, the reviewers agreed that the authors should examine the effects of adding rifampicin on WT-SMC and SMC-CC425 with all parS, parS-9 kb, and two parS (e.g. parS-9 kb and parS-304 kb). The results in Figure 4 with higher levels of WT-SMC could also potentially benefit from the addition of a +rif experiment, though this was deemed lower priority.

3) In some cases, conclusions about the actions of SMC in individual cells are being made based on Hi-C data that intrinsically average the behavior of SMC in large populations of cells. This fact should be more carefully taken into account when interpreting and discussing the data presented.

4) In general, the authors should, in light of the comments from the reviewers ensure that the conclusions they draw take into account the full set of possible models that explain their results, not just the favored model. In other words, if alternative models cannot be fully ruled out in some cases, please make that explicit in the text and tone down conclusions accordingly.

5) Finally, although not noted in the reviews, the discussion among the reviewers led to the suggestion that the paper should better discuss the simulations in the pre-print from Wang and Mirny, which is cursorily discussed in the current version of the manuscript at hand.

*Reviewer #1 (Recommendations for the authors):*

The data generally support the conclusions.

I would like to see inclusion of some chromatin fractionation experiments to directly address the quantity of Condensin in free versus chromosome-associated states – many of the authors conclusions would be strengthened by inclusion of these data.

*Reviewer #2 (Recommendations for the authors):*

Below I list out a few other suggestions/concerns for the authors to consider:

1. Figure 3B: the insertion of two equally strong parS sites on an otherwise parS-less background showed reduced loop formation/arm alignment. This can be attributed to SMC collision or that the total pool of the SMC loader (ParB) is being shared equally between the two parS sites, leading to fewer SMC available to travel from each site. Can the authors rule out the later possibility? I think this is an important point since the accumulation of SMC in the interval between the 2 parS sites (as seen by a population-averaged method such as ChIP-seq, Figure 3C or so) does not necessarily mean that SMCs collide there.

2. ParB can slide at a short distance but might bridge over a longer distance, should the authors consider the possibility that the very weak Hi-C interactions (orange box, Figure 3B ) is due to the bridging of ParB-DNA from the two distal sites

3. How did the authors distinguish experimentally between SMC loading rate and SMC unloading rate? For example, lines 196-199: "..indicating that the modified SMC coiled coil impede DNA translocation and/or increased the rate of unloading". Why an increase in unloading rate? The proposed increase in unloading rate for the arm-modified SMC-CC425 is very counter-intuitive to me when I looked into the magnified ChIP-seq profiles (Figure 2A). Figure 2A (left panel, WT SMC): parS-9kb does not align with the summit of the highest ChIP-seq peak. That is probably due to SMC unloading/escaping from the loading site parS. However, in Figure 2A (right panel, SMC-CC425), parS-9kb aligns perfectly with the summit of the highest ChIP-seq peak. Naively, that suggests to me that SMC-CC425 has a problem either (i) unloading less from parS (rather than more as proposed) or (ii) SMC-CC425 somehow being held on more tightly by the loader ParB. Could the authors clarify this, please?

4. Is this possible, given available data, to speculate on why there is no major arm-alignment in an all-parS + SMC-CC425 strain (Figure 1B)? Is that because of too many collisions in the main parS cluster that cannot be resolved by a defective SMC-CC425, leading to unproductive loop formation?

*Reviewer #3 (Recommendations for the authors):*

Improvements on the presentation of previous models and results from the literature:

– Is there direct evidence that bacterial condensins also work by loop extrusion? This is not clear to me and is an important assumption of the authors.

– "Smc-ScpAB complexes start DNA translocation from predefined entry sites". The manuscript cites Gruber and Errington and Sullivan (2009) but did these studies show that SMC-ScpAB translocate on DNA?

– Citations of ParB binding specifically at *parS* should rather acknowledge the original papers describing this, as it is unimportant whether ParB forms a clamp or not.

– The first to describe condensin-dependent co-alignment of chromosome arms in Bacillus were the papers of Wang et al. (2015) and Marbouty et al. (2015). These should be cited for this instead of: Minnen et al., 2016;Tran et al. 2017; Wang et al., 2017.

– The authors seem to assume in the introduction that Bs SMCs translocates by a two-sided mechanism. But has this been demonstrated?

– Removal of parS/ParB in Bs does have an impact in chromosome conformation and segregation, as shown by Wang et al. (2014). Thus, I would be careful with conclusion that arm-alignment is not required for efficient translocation.

– Wang 2015, and Marbouty 2015 both showed that a single parS site display arm alignment. The authors cite only Wang 2017.

---

## [Author Response]

1) The authors need to include a quantification of SMC (and SMC variant) levels, including an assessment by chromatin fractionation of the quantity free in solution versus chromosome-associated. There are also several cases of quantification of Hi-C data that are needed.

We have now performed additional experiments to estimate the levels of Smc and Smc-CC425 in cell extracts and on the chromosome. Briefly, Western blotting indicated that Smc-CC425 might be less abundant than Smc, however, we believe this might be due to poor blotting (of the larger Smc-CC425 protein) or poor recognition of the chimeric Smc-CC425 protein. Consistent with this notion, we found that a HaloTagged version of the protein exhibited nearly identical signal by in-gel fluorescence detection as wild-type Smc-HaloTag protein (Figure 1—figure supplement 1).

We have also estimated the chromosomal levels of these proteins by the chromosome entrapment assay. The input fractions showed similar levels of cross-linked species, demonstrating that Smc-CC425 assembles normal Smc-ScpAB complexes (Figure 1—figure supplement 1). The eluate fractions showed only slightly reduced signals for Smc-CC425 when compared to wild-type Smc, being consistent with the notion of a mildly reduced chromosome residency time of Smc-CC425 and suggesting that defects in arm alignment in strains with multiple *parS* do not originate from an aberrant failure of loading or entrapping DNA by Smc-CC425.

We also provide quantifications of 3C-seq data, showing that the extent of arm alignment is reduced in all Smc-CCxxx proteins (Figure 1—figure supplement 2A). However, we had difficulties in obtaining trustworthy estimates for the velocity of DNA translocation by Smc-CC425 (from ChIP-seq analyses or by alignment of chromosome arms). We think this is due to elevated spontaneous unloading of Smc-CC425 and thus reduced levels of Smc-CC425 particularly further away from the loading sites. The apparent velocity is thus not reduced at later time points.

2) Additional experiments are needed to help address the relative contributions of SMC-RNAP and SMC-SMC collisions. In particular, the reviewers agreed that the authors should examine the effects of adding rifampicin on WT-SMC and SMC-CC425 with all parS, parS-9 kb, and two parS (e.g. parS-9 kb and parS-304 kb). The results in Figure 4 with higher levels of WT-SMC could also potentially benefit from the addition of a +rif experiment, though this was deemed lower priority.

We have repeated key experiments in the presence of the transcription elongation inhibitor rifampicin (see Figure 1—figure supplement 2; Figure 3—figure supplement 1; Figure 4—figure supplement 1). The results indicate that blockage by transcription indeed contributes to the defects in chromosome organization observed in Smc-CC425. Addition of rifampicin largely eliminates the contact arc formed at ^-304kb^parS and leads to a notable but faint secondary diagonal in Smc-CC425. In contrast, in SMC^high^ cells, contacts formed from ^-304kb^parS are still highly asymmetrical, indicating that Smc-Smc collisions are largely responsible for hindrance of Smc translocation.

3) In some cases, conclusions about the actions of SMC in individual cells are being made based on Hi-C data that intrinsically average the behavior of SMC in large populations of cells. This fact should be more carefully taken into account when interpreting and discussing the data presented.

We have aimed to provide a more careful interpretation of our results following the comments and suggestions below.

4) In general, the authors should, in light of the comments from the reviewers ensure that the conclusions they draw take into account the full set of possible models that explain their results, not just the favored model. In other words, if alternative models cannot be fully ruled out in some cases, please make that explicit in the text and tone down conclusions accordingly.

Similarly, we have aimed to be more balanced and take multiple models into consideration when interpreting our data following the comments and suggestions below.

5) Finally, although not noted in the reviews, the discussion among the reviewers led to the suggestion that the paper should better discuss the simulations in the pre-print from Wang and Mirny, which is cursorily discussed in the current version of the manuscript at hand.

We have now included a slightly more detailed discussion of the simulation manuscript by Brandão et al., 2020.

Reviewer #1 (Recommendations for the authors):The data generally support the conclusions.I would like to see inclusion of some chromatin fractionation experiments to directly address the quantity of Condensin in free versus chromosome-associated states – many of the authors conclusions would be strengthened by inclusion of these data.

As described above, we have now included experiments determining the relative levels of Smc and Smc-CC425 in cell extracts (Western blotting and in-gel fluorescence detection) and on the chromosome (chromosome entrapment assay). We find that cellular and chromosomal levels are roughly comparable (Figure 1—figure supplement 1).

Reviewer #2 (Recommendations for the authors):Below I list out a few other suggestions/concerns for the authors to consider:1. Figure 3B: the insertion of two equally strong parS sites on an otherwise parS-less background showed reduced loop formation/arm alignment. This can be attributed to SMC collision or that the total pool of the SMC loader (ParB) is being shared equally between the two parS sites, leading to fewer SMC available to travel from each site. Can the authors rule out the later possibility? I think this is an important point since the accumulation of SMC in the interval between the 2 parS sites (as seen by a population-averaged method such as ChIP-seq, Figure 3C or so) does not necessarily mean that SMCs collide there.

We believe this comment is particularly relevant for Figure 4G. We have added the following statement to the Results section: ‘However, we cannot rule out that this effect is mainly caused by competition between the two *parS* sites for Smc loading.’ (Please note that the quality of these maps has been improved by additional DNA sequencing).

Several findings indicate that ParB levels at *parS* are not rate-limiting for Smc loading. For example, if ParB-parS clusters were limiting Smc loading, then in SMC^high^ one would expect a similar outcome as with wild-type Smc levels. In a recent preprint (Antar et al., 2021) we also showed that Smc is efficiently recruited by limited amounts of (a mutant of) ParB. ParB sharing by multiple *parS* sites thus does not seem to be a limiting factor for Smc loading and function (translocation/ arm alignment). Moreover, we do not see how a reduction in loading at one *parS* site would lead to increased levels between the two *parS* sites.

2. ParB can slide at a short distance but might bridge over a longer distance, should the authors consider the possibility that the very weak Hi-C interactions (orange box, Figure 3B ) is due to the bridging of ParB-DNA from the two distal sites

ParB has indeed been suggested to organize DNA by DNA bridging in the vicinity of *parS*. However, ParB occupancy of *parS* flanking DNA is limited to tens of kb. Here, we observed longer-distance (>100 kb) interactions that seem to be out of the range of ParB-mediated DNA contacts.

3. How did the authors distinguish experimentally between SMC loading rate and SMC unloading rate? For example, lines 196-199: "..indicating that the modified SMC coiled coil impede DNA translocation and/or increased the rate of unloading". Why an increase in unloading rate? The proposed increase in unloading rate for the arm-modified SMC-CC425 is very counter-intuitive to me when I looked into the magnified ChIP-seq profiles (Figure 2A). Figure 2A (left panel, WT SMC): parS-9kb does not align with the summit of the highest ChIP-seq peak. That is probably due to SMC unloading/escaping from the loading site parS. However, in Figure 2A (right panel, SMC-CC425), parS-9kb aligns perfectly with the summit of the highest ChIP-seq peak. Naively, that suggests to me that SMC-CC425 has a problem either (i) unloading less from parS (rather than more as proposed) or (ii) SMC-CC425 somehow being held on more tightly by the loader ParB. Could the authors clarify this, please?

We have not been able to directly measure Smc loading and unloading rates. However, the newly added chromosome co-entrapment assay indicates that Smc-CC425 is loaded slightly less well onto the chromosome than wild-type Smc rather than better. This supports our initial interpretation.

Given the finite number of Smc complexes in the cell (Wilhelm et al., 2015) and the expected high residency time of the complexes on the chromosome (Banigan et al., 2019), we see that the initial high enrichment near the replication origin in t_20_ after ParB repletion is promptly followed by a reduction, suggesting that most complexes are now associated with the chromosome and are not available for loading (Figure 2C). Most of them will only become available again for re-loading after dissociating from the chromosome near *ter* (Karaboja et al., 2021). In case of Smc-CC425, the complexes remain enriched in the replication origin region. Since the velocity of Smc-CC425 and its entrapment capacity appear only partially affected compared to wt Smc, it implies that the Smc-CC425 enrichment in the origin region is most likely due to spontaneous unloading during translocation and reloading in the origin region.

4. Is this possible, given available data, to speculate on why there is no major arm-alignment in an all-parS + SMC-CC425 strain (Figure 1B)? Is that because of too many collisions in the main parS cluster that cannot be resolved by a defective SMC-CC425, leading to unproductive loop formation?

Yes, based on our experiments, we suggest that an increased flux of Smc-CC425 loading onto the chromosome at the cluster of main *parS* sites (parS-354, 355, 356 and 359) (and a possibly reduced translocation rate) lead to frequent Smc collisions and unproductive loop formation. Conversely, Smc loading at ^—304kb^parS site, due to its more remote position, has a particularly high chance of forming a noticeable DNA loop before encountering another Smc acting upon a different parS site/sites.

To underscore this point, we have added following statement to the relevant section of results:

“We suggest that hyper-enrichment of Smc-CC425 leads to frequent collisions and unproductive loop formation.”

Reviewer #3 (Recommendations for the authors):Improvements on the presentation of previous models and results from the literature:– Is there direct evidence that bacterial condensins also work by loop extrusion? This is not clear to me and is an important assumption of the authors.

Bacterial Smc-ScpAB complexes are known to be loaded at parS sites and then translocate bidirectionally away from parS. This process is equivalent to the process of DNA loop extrusion in eukaryotes (albeit with predefined loading rather than stalling sites). To highlight the equivalence of the SMC processes in eukaryotes and prokaryotes, we find it useful to use the same term to describe SMC activity also in bacteria.

– "Smc-ScpAB complexes start DNA translocation from predefined entry sites". The manuscript cites Gruber and Errington and Sullivan (2009) but did these studies show that SMC-ScpAB translocate on DNA?

Thanks. We have added additional citations to support this statement (Minnen et al., 2016; Wang et al., 2017).

– Citations of ParB binding specifically at parS should rather acknowledge the original papers describing this, as it is unimportant whether ParB forms a clamp or not.

Thanks. We have added the original citation (Lin and Grossman, 1998) and revised the sentence.

– The first to describe condensin-dependent co-alignment of chromosome arms in Bacillus were the papers of Wang et al. (2015) and Marbouty et al. (2015). These should be cited for this instead of: Minnen et al., 2016; Tran et al. 2017; Wang et al., 2017.

Thanks. We have added the citations.

– The authors seem to assume in the introduction that Bs SMCs translocates by a two-sided mechanism. But has this been demonstrated?

Unidirectional translocation does not explain the available data (3C-seq and ChIP-seq time course), such as chromosome folding with *parS* being the inflection point. While not formally proven, it seems beyond reasonable doubt to assume a two-sided translocation mechanism (by a single or a double SMC complex).

– Removal of parS/ParB in Bs does have an impact in chromosome conformation and segregation, as shown by Wang et al. (2014). Thus, I would be careful with conclusion that arm-alignment is not required for efficient translocation.

Chromosome segregation can happen efficiently with little or no evidence of arm-alignment (in population-averaged 3C-seq maps) as observed in multiple mutants. Complete removal of *parS* and ParB, however, has clear impacts on chromosome segregation fidelity (mostly by hindering Smc loading and maybe also by eliminating ParABS function).

We have tried to make this point clearer in the manuscript.

– Wang 2015, and Marbouty 2015 both showed that a single parS site display arm alignment. The authors cite only Wang 2017.

Added to the manuscript (Marbouty et al., 2015; Wang et al., 2015)